Manuscript prepared for Atmos. Chem. Phys.
with version 2014/09/16 7.15 Copernicus papers of the LATEX class copernicus.cls.
Date: 20 July 2018

# Changes in the aerosol direct radiative forcing from 2001 to 2015: observational constraints and regional mechanisms

Fabien Paulot[1,2], David Paynter[1], Paul Ginoux[1], Vaishali Naik[1], and Larry W. Horowitz[1]

[1]Geophysical Fluid Dynamics Laboratory, National Oceanic and Atmospheric Administration, Princeton, New Jersey, USA
[2]Program in Atmospheric and Oceanic Sciences, Princeton University, New Jersey, USA

*Correspondence to:* Fabien.Paulot@noaa.gov

**Abstract.** We present observation- and model-based estimates of changes in the aerosol direct clear-sky shortwave radiative effect ($\mathrm{DRE_{clr}^{sw}}$), the perturbation by aerosols of the net downward shortwave clear-sky radiation at the top of the atmosphere. Observation-based estimates of $\mathrm{DRE_{clr}^{sw}}$ are derived from the outgoing shortwave clear-sky radiation measured by the Clouds and the Earth's Radiant Energy System (CERES) accounting for the effect of variability in surface albedo, water vapor, and ozone. From 2001 to 2015, we find that $\mathrm{DRE_{clr}^{sw}}$ increases (i.e., less radiation is scattered to space by aerosols) over Western Europe ($0.7 - 1 \ \mathrm{W\,m^{-2}\,decade^{-1}}$) and the Eastern US ($0.9 - 1.8 \ \mathrm{W\,m^{-2}\,decade^{-1}}$), decreases over India ($-0.5 - -1.9 \ \mathrm{W\,m^{-2}\,decade^{-1}}$), and does not change significantly over Eastern China. We show that the GFDL chemistry-climate model AM3, driven by CMIP6 historical emissions, captures changes over Western Europe ($0.6 \ \mathrm{W\,m^{-2}\,decade^{-1}}$) and the Eastern US ($0.8 \ \mathrm{W\,m^{-2}\,decade^{-1}}$) well. This agreement reflects the mature understanding of the sulfate budget in these regions. In contrast, the model overestimates the change in $\mathrm{DRE_{clr}^{sw}}$ over India and Eastern China. Over China, this bias can be partly attributed to the decline of $SO_2$ emissions after 2007, which is not captured in the CMIP6 emission inventory. Over India, the lack of change in the outgoing clear-sky shortwave radiation in the model is attributed to the compensation between the decreases of dust and surface albedo and the increase of anthropogenic aerosols. For both India and Eastern China, model simulations indicate that nitrate and black carbon contribute more to changes in $\mathrm{DRE_{clr}^{sw}}$ than in the US and Europe, which highlights the need to better constrain their sources and chemistry. Globally, our model shows that changes in the all-sky aerosol direct radiative forcing, the anthropogenic component of the aerosol direct radiative effect, between 2001 and 2015 ($+0.03 \ \mathrm{W\,m^{-2}}$) are dominated by black carbon ($+0.12 \ \mathrm{W\,m^{-2}}$) with significant offsets from nitrate ($-0.03$

$W\,m^{-2}$) and sulfate (-0.03 $W\,m^{-2}$). AM3 also shows that changes in the speciation and spatial distribution of emissions between 2001 and 2015 have reduced the sensitivity of the aerosol direct radiative forcing to $SO_2$ emissions (via sulfate aerosols), but increased that to ammonia and NO emissions (via nitrate aerosols).

## 1 Introduction

Aerosols affect climate (Boucher et al., 2013) both directly, via scattering and absorption of solar and terrestrial radiation (Charlson et al., 1992), and indirectly, by modulating the abundance of cloud condensation nuclei, the droplet size distribution, and the lifetime of clouds (Twomey, 1974; Rosenfeld et al., 2014). Storelvmo et al. (2016) estimated that the increase in the burden of atmospheric aerosols associated with anthropogenic activities has masked approximately one-third of the continental warming from greenhouse gases from 1964 to 2010, with important implications for global and regional climate (Wild, 2009; Bollasina et al., 2011).

Previous studies have leveraged global spaceborne observations of the Earth's radiative budget (Wielicki et al., 1996, 1998) and aerosol abundance (Kahn et al., 2005; Levy et al., 2013b) to estimate the overall aerosol direct radiative effect (DRE), i.e., the direct perturbation of the Earth's radiative budget by aerosols (Christopher and Zhang, 2004; Patadia et al., 2008; Loeb and Manalo-Smith, 2005; Kahn, 2012). Observational constraints for the aerosol direct radiative forcing (DRF), the anthropogenic component of the aerosol direct radiative effect, are less robust (Su et al., 2013; Bellouin et al., 2005, 2008), which contributes to the large spread in model estimates for DRF in 2000 relative to 1850 (-0.02 – -0.58 $W\,m^{-2}$ Myhre et al. (2013)). In particular, the sensitivity of the aerosol direct radiative forcing to anthropogenic emissions remains uncertain. Previous work has shown that the aerosol forcing simulated by global climate models from 1850 to 2001 is well correlated with changes in $SO_2$ emissions (Stevens, 2015). However, this relationship may not be applicable in recent years and for future conditions, as the spatial distribution and speciation of anthropogenic emissions evolve (Stevens et al., 2017).

In this work, we aim to provide observational constraints on the sensitivity of the direct aerosol forcing to anthropogenic emissions. The paper is organized as follows: First, we derive an estimate of changes in the clear-sky shortwave aerosol direct radiative effect from 2001 to 2015 constrained by the observed variability in outgoing shortwave radiation from the Clouds and the Earth's Radiant Energy System (CERES). Second, we focus on large source regions of anthropogenic emissions (US, Europe, India, and Eastern China), where observed changes in the aerosol effect are expected to be dominated by anthropogenic aerosols. This allows us to assess whether a state-of-the-art chemistry-climate model (Geophysical Fluid Dynamics Laboratory (GFDL) AM3) driven by the latest emissions from the Coupled Model Intercomparison Project phase 6 can capture changes in the direct radiative forcing from aerosols over the 2001–2015 period. Finally we use AM3 to compare the

sensitivity of the aerosol direct radiative forcing to anthropogenic emissions from 1850 to 2001 and from 2001 to 2015.

## 2    Methods

### 2.1    GFDL-AM3 model

We use the GFDL-AM3 model (Donner et al., 2011; Naik et al., 2013), the atmospheric chemistry climate component of the GFDL-CM3 model (Donner et al., 2011; Griffies et al., 2011; John et al., 2012). The model is run from 2000 to 2015, using the first year to spin up the model. The model horizontal resolution is $\simeq$ 200 km with 48 vertical levels. To facilitate comparisons with synoptic observations, the model horizontal winds are nudged to 6-hourly horizontal winds from the National Centers for Environmental Predication reanalysis (Kalnay et al., 1996). Monthly sea surface temperature and sea ice concentration are prescribed following Taylor et al. (2000) and Rayner et al. (2003), respectively. The configuration of AM3 used in this study includes revisions to the representation of the wet scavenging of chemical tracers by snow and convective precipitation and to the treatment of sulfate and nitrate chemistry, which significantly improve the representation of aerosols. We refer the reader to our recent work for a detailed evaluation of the aerosol simulation in AM3 (Paulot et al., 2016).

The radiative transfer scheme takes into account the aerosol optical properties of sulfate, sea salt, dust, black carbon, organic carbon (Donner et al., 2011) and nitrate (Paulot et al., 2017b). Aerosols are assumed to be externally mixed, except for hydrophilic black carbon and sulfate (Donner et al., 2011). Hygroscopic growth is capped at 95% for all aerosols.

We use the historical anthropogenic emissions developed by the Community Emission Data System (CEDS v2017-05-18) in support of CMIP6 (Hoesly et al., 2018). As anthropogenic emissions are available until 2014 from CEDS, we repeat CEDS 2014 anthropogenic emissions for 2015. Monthly biomass burning emissions are from the historical global biomass burning emissions inventory for CMIP6 (BB4CMIP6, van Marle et al. (2017)). Emissions for the 1997 to 2015 period in this inventory have been derived from satellite-based emissions from the Global Fire Emissions Database (GFED, van der Werf et al. (2017)). The vertical distribution of biomass burning emissions is taken from Dentener et al. (2006). Natural emissions are based on Naik et al. (2013), except for isoprene emissions, which are calculated interactively using the Model of Emissions of Gases and Aerosols from Nature (MEGAN, Guenther et al. (2006)).

Fig. 1 shows changes in the anthropogenic emissions of sulfur dioxide ($SO_2$), ammonia ($NH_3$), black carbon (BC), and nitrogen oxide (NO) from 2001 to 2015. Globally, anthropogenic emissions of $NH_3$, BC, and NO have increased by 18%, 36%, and 16% over the 2001-2015 period, respectively, while $SO_2$ emissions have remained nearly stable, peaking in 2006. In the US and Europe, there have been significant declines in $SO_2$ (-71% and -66%, respectively) and NO (-48% and -39%) emissions,

while $NH_3$ and BC emissions have changed little (<15%). Indian emissions of $SO_2$, NO, and BC have increased by 89%, 39%, and 89%. Similarly, Chinese emissions of $SO_2$, NO, and BC have increased by 56%, 69%, and 93%, respectively. Anthropogenic emissions in India and China are expected to be more uncertain than in the US and Europe (Saikawa et al., 2017a, b). For instance, Fig. 1 shows differences between emissions from CMIP6 and emissions from the regional Modular Emission Inventory for China (MEIC) (Zhang et al., 2009). Unlike in CMIP6, emissions of $SO_2$ decline starting in 2006, a decrease that accelerates in 2012, while NO emissions decrease after 2012 and BC emissions remain near-stable after 2007. In 2014, MEIC NO, $SO_2$, and BC emissions are 24%, 48%, and 32% lower than CMIP6 emissions, respectively. $NH_3$ emissions are similar in magnitude but exhibit different seasonality: CMIP6 $NH_3$ emissions peak in spring, while MEIC exhibits a broad peak in summer, consistent with top-down constraints (Paulot et al., 2014; Zhang et al., 2017). The impact of these emission uncertainties on the simulated change in the aerosol effect over India and China will be discussed in sections 3.2.2 and 3.2.3, respectively.

## 2.2   Aerosol direct effect and forcing

The instantaneous aerosol direct radiative effect (DRE) is defined as the difference between the outgoing radiation at the top of the atmosphere (TOA) in the absence and in the presence of aerosols (Heald et al., 2014). The direct radiative forcing (DRF) is defined as the anthropogenic component of the direct radiative effect. In our notation we use the superscript $sw$ to denote the shortwave component of DRE or DRF. Likewise, the subscript $clr$ denotes the clear-sky component of DRE or DRF.

To better isolate the effect of aerosol variability on radiative fluxes, we will focus on the aerosol shortwave direct radiative effect under clear-sky conditions ($DRE_{clr}^{sw}$):

$$DRE_{clr}^{sw} = rsutcsaf - rsutcs \tag{1}$$

where we use the CMIP6 convention (CMIP6 Data Request, 2018) to designate the outgoing clear-sky shortwave radiation with (rsutcs) and without aerosols (rsutcsaf), respectively. For simplicity, we will refer to the aerosol shortwave direct radiative effect under clear-sky conditions ($DRE_{clr}^{sw}$) as the aerosol effect, hereafter. Note that an increase of the aerosol direct effect indicates a decrease of the radiation scattered to space by aerosols.

### 2.2.1   Model

In AM3, the aerosol effect is estimated by calling the radiative transfer scheme twice, with and without aerosols (Paulot et al., 2017b) in the absence of clouds. The effect of individual aerosol components is estimated as the difference in outgoing shortwave radiation with and without aerosol $x$, where $x$ can be sulfate, nitrate, black carbon, organic carbon, dust, sea salt, and stratospheric volcanic aerosols. In the following, we will focus primarily on changes in sulfate and nitrate, which

dominate changes in aerosol scattering, and black carbon, which dominates changes in aerosol absorption over the 2001–2015 period.

### 2.2.2 Observations

The Clouds and the Earth's Radiant Energy System (CERES, Wielicki et al. (1996, 1998)) provides constraints on the Earth's radiative budget since 2000. Here, we use the Energy Balanced and Filled product (EBAF, edition 4, Loeb et al. (2018)) to estimate the variability of the clear-sky shortwave outgoing radiation. This product achieves global coverage by combining CERES broadband cloud-free fluxes with MODIS (Moderate Resolution Imaging Spectroradiometer) radiances for regions that are not completely cloud-free at the CERES footprint scale (Loeb et al., 2018).

The simplest way in which CERES EBAF data can be used to estimate changes in the aerosol effect is to assume that all variability in the shortwave clear-sky outgoing radiation is the result of changes in aerosols (Stevens and Schwartz, 2012; Xing et al., 2015; Alfaro-Contreras et al., 2017). We will refer to this estimate as $EBAF_R$ hereafter, where R stands for raw. However, other radiative components may contribute to the variability in the outgoing radiation (Stevens and Schwartz, 2012). Therefore, a more accurate estimate of the aerosol effect requires removal of the impact of these components from the measured changes in the outgoing radiation. To achieve this, we calculate radiative kernels (e.g., Soden et al. (2008); Shell et al. (2008)) to estimate the variability of the outgoing clear-sky shortwave radiation associated with changes in surface albedo, ozone, and water vapor (see supporting materials and Fig. S1). For water vapor and ozone, we use the Goddard Modeling and Assimilation Office reanalysis (GEOS5). Since our estimate for the aerosol effect is most sensitive to changes in the surface albedo, we will consider both the albedo from MODIS (Schaaf et al., 2002) and CERES-EBAF (Rutan et al., 2009, 2015; Loeb et al., 2018). Both albedo estimates have been validated extensively and generally show good agreement with observations (Cescatti et al., 2012; Wang et al., 2014b; Rutan et al., 2009, 2015). Estimates of the aerosol effect derived using the MODIS and CERES-EBAF albedo will be referred to as $EBAF_M$ and $EBAF_C$, respectively.

We also use estimates of changes in the aerosol effect provided by the CERES Synoptic Radiative Fluxes product (SYN, edition 4a). The CERES SYN product calculates fluxes at the top of the atmosphere using a radiative transfer code constrained by observations. These calculations are performed with and without aerosols present, allowing for an estimate of the aerosol effect. The aerosol properties used in the SYN calculations come from the Model for Atmospheric Transport and Chemistry (MATCH) that is constrained by observations from MODIS collection 5 (Collins et al., 2001). Therefore, the SYN calculated aerosol effect is very sensitive to MODIS collection 5 aerosol properties. This collection has now been superseded by MODIS collection 6 (Levy et al., 2013b) and we will discuss some of the implications of differences between MODIS collections 5 and 6 for the derivation of the SYN aerosol effect in sections 3.2.1 and 3.2.2.

### 2.3 Trend: estimation and interpretation

We use the non-parametric Mann-Kendall test (Kendall, 1938) to identify significant changes in the aerosol effect. This test quantifies monotonic correlations between two variables. It is based on a rank procedure that makes it less susceptible to outliers than the Pearson correlation and thus especially well-suited for the analysis of environmental dataset. We estimate the linear trend using the Theil-Sen method (Theil, 1950; Sen, 1968). We use a critical $p$ value of 0.05 for trend significance.

Differences between observed and simulated trends in $\mathrm{DRE}_{\mathrm{clr}}^{\mathrm{sw}}$ may reflect biases in the simulated change of the aerosol burden. Here this is diagnosed by comparing the simulated trend in aerosol optical depth (AOD) with those retrieved by the Multi-angle Imaging SpectroRadiometer (MISR) at 555nm (Kahn et al., 2005, 2010) and the MODIS instruments on board the AQUA and TERRA satellites at 550 nm (collection 6, level3, merged deep blue/dark target) (Levy et al., 2013a; Sayer et al., 2014). Note that the accuracy of individual retrievals has been estimated to be $\pm 0.05 \pm 0.15 \times$ AOD (Levy et al., 2010) for MODIS and the maximum of $\pm 0.05$ or $0.2 \times$ AOD for MISR (Kahn et al., 2010).

The change in AOD is not a perfect predictor of changes in $\mathrm{DRE}_{\mathrm{sw}}^{\mathrm{clr}}$ and we will show that it is possible to find regions where observed changes in AOD are well captured by AM3 but not changes in $\mathrm{DRE}_{\mathrm{sw}}^{\mathrm{clr}}$ (see 3.2.2). Such discrepancies may reflect differences in aerosol radiative properties. Specifically, changes in absorbing aerosols, such as black carbon, have a small imprint on AOD but a large one on $\mathrm{DRE}_{\mathrm{clr}}^{\mathrm{sw}}$ (see section 3.2.2 and 3.2.3). Differences in surface properties may also cause differences in $\mathrm{DRE}_{\mathrm{clr}}^{\mathrm{sw}}$ trends. For instance, a lower surface albedo reduces the impact of changes in scattering aerosols on $\mathrm{DRE}_{\mathrm{sw}}^{\mathrm{clr}}$ and conversely increases that of absorbing aerosols. We will show that such differences in surface albedo are important in India (section 3.2.2). However, in other regions, we find they have a small impact on the simulated trend in $\mathrm{DRE}_{\mathrm{sw}}^{\mathrm{clr}}$.

## 3 Results

In this section, we will refer to the clear-sky shortwave outgoing radiation (rsutcs) and the aerosol shortwave direct radiative effect under clear-sky conditions ($\mathrm{DRE}_{\mathrm{clr}}^{\mathrm{sw}}$) as outgoing radiation and aerosol effect, respectively.

### 3.1 Global distribution of changes in aerosol effect

Fig. 2 shows the decadal rate of change in the aerosol effect, estimated solely from changes in the outgoing radiation ($\mathrm{EBAF}_{\mathrm{R}}$) measured by CERES EBAF (top panel) over the 2001–2015 period. We find significant changes (highlighted with dots) in the outflow of the Eastern US, where the radiation scattered back to space by aerosols decreases, and in the outflow of India, where it increases, consistent with the changes in anthropogenic emissions shown in Fig. 1. However, changes

in the outgoing radiation are less significant over the source regions themselves, which highlights the importance of other factors of variability in the outgoing radiation (Stevens and Schwartz, 2012).

Fig. 2 also shows the decadal rate of change in the aerosol effect derived from the SYN calculation and from CERES-EBAF outgoing radiation after correction for water, ozone, and surface albedo from MODIS ($EBAF_M$) and CERES-EBAF ($EBAF_C$). All these estimates show better spatial consistency between land and ocean near large sources of anthropogenic pollution than the outgoing radiation alone ($EBAF_R$). In particular, we find that the aerosol effect increases over North America and Europe, and decreases over India. In contrast, the variability is considerably reduced over Australia, Central Asia, and South America, which suggests that it is not primarily associated with aerosols. Consistent with observations, AM3 also shows that the aerosol effect increases over the US and Europe and decreases over India. However, it simulates a decrease in the aerosol effect over China and in the Western Pacific, which is inconsistent with observational constraints.

To understand these changes further, we examine the timeseries of the different estimates of the aerosol effect over these regions. $EBAF_R$ exhibits considerable interannual variability over the Eastern US and Europe, with no significant trend (Table S1). In contrast, SYN, $EBAF_C$, and $EBAF_M$ estimates exhibit a significant increase ranging from 0.9 to 1.8 $\mathrm{W\,m^{-2}\,decade^{-1}}$ in the Eastern US and from 0.7 to 1.4 $\mathrm{W\,m^{-2}\,decade^{-1}}$ in Western Europe. AM3 also simulates an increase of the aerosol effect over these regions (0.8 and 0.6 $\mathrm{W\,m^{-2}\,decade^{-1}}$, respectively). The magnitude of these changes is in excellent agreement with $EBAF_M$ but lower than SYN. We refer the reader to section 3.2.1 for a detailed discussion of these regions.

Over India, most observational estimates (SYN, $EBAF_C$, $EBAF_M$) suggest a decrease of the aerosol effect ($-1.0 - -1.9\,\mathrm{W\,m^{-2}\,decade^{-1}}$), which is qualitatively captured by AM3 (-2.4 $\mathrm{W\,m^{-2}\,decade^{-1}}$). However, changes in the outgoing radiation alone ($EBAF_R$) would imply a small increase of the aerosol effect from 2001 to 2015 (0.5 $\mathrm{W\,m^{-2}\,decade^{-1}}$), which suggests that large changes in other radiative components may be masking the aerosol effect. Changes in the aerosol effect over India will be discussed in section 3.2.2.

Over Eastern China, all observational estimates of the aerosol effect exhibit a rapid decrease from 2001 to 2007, followed by an increase until 2015, with no significant trend overall in SYN, $EBAF_C$, and $EBAF_M$. The timing of the reversal is consistent with previous analysis of changes in AOD (Zhao et al., 2017) and outgoing radiation over the China sea (Alfaro-Contreras et al., 2017). AM3 fails to capture this reversal and simulates a significant decrease in the aerosol effect from 2001 to 2015 (-1.3 $\mathrm{W\,m^{-2}\,decade^{-1}}$). Changes in the aerosol effect over China will be discussed in section 3.2.3.

We note that all observation-based estimates of DRE show some significant changes over remote oceanic regions. These changes may reflect cloud contaminations in the CERES cloud filtering algorithm (for $EBAF_C$, $EBAF_M$) and in the aerosol retrieval (SYN). In addition, low aerosol loadings make $EBAF_C$ and $EBAF_M$ more susceptible to errors in the radiative kernels.

### 3.2 Regional changes

#### 3.2.1 Western Europe and Eastern US

Fig. 4 (top row) shows the seasonal change of the AOD and aerosol effect over Europe. Observations show that the AOD decreases most in spring and summer (-0.4 $dec^{-1}$ for MODIS TERRA (solid black line), Table 1). This decrease is accompanied by an increase of the aerosol effect of 1–1.8 $W\,m^{-2}\,decade^{-1}$ in spring and 1.2–2.5 $W\,m^{-2}\,decade^{-1}$ in summer (Fig. 4 (bottom row) and Table 1). AM3 captures these changes well (Table S2). In the model, both changes in AOD and aerosol effect are driven almost entirely by the decrease of sulfate aerosols associated with the decrease of $SO_2$ emissions. The slower changes in winter and fall reflects the smaller contribution of sulfate to the aerosol burden and the less efficient oxidation of $SO_2$ in these seasons, which makes sulfate less sensitive to changes in $SO_2$ emissions (Wang et al., 2011; Paulot et al., 2017a).

Fig. 5 shows the changes of the AOD and aerosol effect over the Eastern US. The overall pattern is similar to Western Europe with large reductions in AOD (up to -0.11 $dec^{-1}$) and increases in the aerosol effect (up to 3.6 $W\,m^{-2}\,decade^{-1}$) in spring and summer (Table 1). AM3 underestimates MODIS AOD as well as the rate of change of the AOD and aerosol effect in summer (Table 1). This is consistent with the model low bias against sulfate concentration in rain water in the US (Paulot et al., 2016). Similar to observations, AM3 also shows greater seasonal contrast between spring and summer in the US than in Europe. In the model, this is driven by more efficient springtime oxidation of $SO_2$ in Europe, where high emissions of ammonia facilitate its in-cloud oxidation by ozone (Paulot et al., 2017a).

In both Europe and the US, we find that the change in the aerosol effect inferred from the SYN calculation is larger than that estimated from CERES-EBAF outgoing radiation corrected for surface albedo changes ($EBAF_C$ and $EBAF_M$). The magnitude of the changes in the MATCH AOD, which is based on MODIS collection 5 and used to calculated the SYN aerosol effect, is also greater than that inferred from the improved MODIS collection 6 (Table 1). This suggests that the rate of change in SYN aerosol effect may be biased high in Europe and Western Europe.

#### 3.2.2 India

Fig. 6 shows the changes in AOD and aerosol effect over India. We will focus here on changes during the winter (DJF) and premonsoon seasons (MAM).

Previous studies have shown that aerosols are primarily of anthropogenic origin in winter (Babu et al., 2013; Pan et al., 2015). During this season, all instruments show a significant increase in AOD (up to 0.13 $dec^{-1}$). In spite of this increase, the outgoing radiation ($EBAF_R$) does not exhibit a significant trend. We attribute this apparent inconsistency to a concurrent decrease in surface albedo (Table S3), which may be associated with the increase in the regional greenness leaf area index reported by Zhu et al. (2016). Accounting for changes in surface albedo, we diagnose a decrease

in the aerosol effect ranging from -0.8 $\mathrm{W\,m^{-2}\,decade^{-1}}$ (using MODIS albedo, $\mathrm{EBAF_M}$) to -2.3 $\mathrm{W\,m^{-2}\,decade^{-1}}$ (using CERES-EBAF albedo, $\mathrm{EBAF_C}$). The large difference between $\mathrm{EBAF_C}$ and $\mathrm{EBAF_M}$ reflects the difference between MODIS and CERES-EBAF albedo in this region (Table S3).

Fig. 6 shows that the simulated AOD agrees well for both magnitude and trend with MODIS AOD but overestimates the change in MISR AOD (see Table 1). The simulated change in the aerosol effect (-2.7 $\mathrm{W\,m^{-2}\,decade^{-1}}$) agrees well with the $\mathrm{EBAF_C}$ and SYN estimates (-2.3 and -2.6 $\mathrm{W\,m^{-2}\,decade^{-1}}$, respectively). However, this good agreement is fortuitous, as the higher surface albedo in AM3 (0.166) relative to SYN (0.129) or CERES-EBAF (0.135) tends to dampen changes

in the simulated aerosol scattering. Specifically, we estimate that the simulated trend in the aerosol effect would be -3.5 $\mathrm{W\,m^{-2}\,decade^{-1}}$ if AM3 was forced with the SYN albedo. This suggests that AM3 overestimates the decrease in the aerosol effect by 1 to 2 $\mathrm{W\,m^{-2}\,decade^{-1}}$. Many factors could contribute to this bias. Here we focus on the seasonality of the emissions of black carbon and ammonium nitrate precursors. Black carbon is the largest contributor to aerosol absorption over In-

dia (+3.2 $\mathrm{W\,m^{-2}}$ on average in winter). Its increase cancels out one third (0.9 $\mathrm{W\,m^{-2}\,decade^{-1}}$) of the decrease in the aerosol effect, much more than in the US and Europe. This is likely to be an underestimate as the prevalent use of biofuel in winter for heating, a large source of black carbon (Yevich and Logan, 2003; Pan et al., 2015), is not represented in the CMIP6 emission inventory. Nitrate dominates changes in the aerosol scattering in winter (-2.4 $\mathrm{W\,m^{-2}\,decade^{-1}}$). This is con-

sistent with previous multi-model assessments, which showed that models that did not include nitrate severely underestimated the AOD over India (Pan et al., 2015). Nitrate is formed via the reaction of ammonia (primarily from agriculture) and nitric acid (from the oxidation of NO, whose emissions are dominated by fossil fuel combustion). Nitrate remains challenging to represent in models because of uncertainties in both ammonia emissions and its chemistry and removal (Heald et al., 2012;

Paulot et al., 2016). In particular, the seasonality of Indian ammonia emissions in CMIP6 is based on European emissions and peak in spring. In contrast, Warner et al. (2017) recently showed that the ammonia column peaks in summer over India (Fig. S2). Using AM3, we estimate that modulating ammonia emissions with the seasonality derived from satellite would reduce the simulated trend in the aerosol effect in winter from -2.7 to -1.9 $\mathrm{W\,m^{-2}\,decade^{-1}}$. These suggest that uncer-

tainties in the seasonalities of black-carbon and ammonia emissions alone could explain most of the discrepancy between observed and simulated changes in the wintertime aerosol effect.

In the premonsoon season, the AOD changes much less rapidly than in winter (Fig. 6, Table 1). For instance, MODIS (TERRA) AOD increases by 0.04 $\mathrm{decade^{-1}}$, less than a third of the rate in winter. This seasonal contrast is not captured by AM3, which simulates a similar change (0.15 $\mathrm{decade^{-1}}$) in

both seasons (Table 1). This discrepancy can be partly explained by the decrease of dust optical depth (dash black line, -0.07 $\mathrm{decade^{-1}}$) diagnosed from MODIS following Ginoux et al. (2012). This decline, which is not captured by AM3, is supported by the decline of coarse-mode aerosols in the

Indo-Gangetic Plain (Babu et al., 2013). Using the simulated relationship between dust optical depth and dust aerosol effect, we estimate that the reduction in dust optical depth has caused an increase in the aerosol effect of 1.4 $\mathrm{W\,m^{-2}\,decade^{-1}}$. This suggests that the decline of dust accounts for most of the discrepancy between the model (-3.1 $\mathrm{W\,m^{-2}\,decade^{-1}}$) and the observational estimates of changes in the aerosol effect (-0.9 – -1.4 $\mathrm{W\,m^{-2}\,decade^{-1}}$).

Jin and Wang (2018) recently suggested that an increase in rainfall in Northwestern India has caused a regional greening, which has been accompanied by a reduction of dust emissions. This mechanism may explain why the Goddard Chemistry Aerosol Radiation and Transport (GOCART), which includes the modulation of dust emissions by LAI (Kim et al., 2013), captures the decrease of dust in this region (Babu et al., 2013). This suggests that the impact of increasing anthropogenic aerosols on the outgoing radiation may have been masked by regional greening both directly (via the decrease of the surface albedo) and indirectly (via lower dust emissions).

### 3.2.3 Eastern China

Fig. 7 shows the change in AOD and aerosol effect over Eastern China. AM3 captures the average magnitude of AOD well in winter and spring but underestimates AOD (MODIS) during the monsoon and post monsoon seasons (Table 1). Although there are significant differences between the different AOD retrievals (Zhao et al., 2017), no significant trend is detected in either AOD or aerosol effect over the entire 2001-2015 period in any season.

In contrast to observations, simulated AOD and aerosol effect exhibit significant changes in both spring (0.15 $\mathrm{decade^{-1}}$ and -2.1 $\mathrm{W\,m^{-2}\,decade^{-1}}$, respectively) and summer (0.11 $\mathrm{decade^{-1}}$ and -1 $\mathrm{W\,m^{-2}\,decade^{-1}}$, respectively). In spring, sulfate is the largest contributor to the AOD and aerosol effect but changes are dominated by nitrate aerosols (0.08 $\mathrm{decade^{-1}}$ and -2.2 $\mathrm{W\,m^{-2}\,decade^{-1}}$, respectively (Table 1). This large springtime change in nitrate is associated with the May maximum of ammonia emissions in the CMIP6 emission inventory.

Similar to India, the model bias may be associated with uncertainties in anthropogenic emissions. As noted in section 2.1, there are significant differences between the CMIP6 and MEIC emission inventories for $SO_2$ after 2007 and NO after 2013 (Fig. 1). A detailed evaluation of these two emission inventories is beyond the scope of this study. However, observations show significant declines in $SO_2$ columns starting in 2008 (Li et al., 2010; Irie et al., 2016; de Foy et al., 2016; Liu et al., 2016; Ding et al., 2017; van der A et al., 2017; Krotkov et al., 2016) and $NO_2$ starting in 2012 (Liu et al., 2016; van der A et al., 2017), consistent with MEIC emissions. We refer the reader to the study of van der A et al. (2017) for a detailed discussion of the technological and regulatory changes that have contributed to the changes in Chinese emissions over the 2001-2015 period.

To quantify the sensitivity of our results to these uncertainties, we perform another simulation replacing the CMIP6 emission by the MEIC emissions for NO, BC, $SO_2$, and $NH_3$ over China. We find that the reduction of $SO_2$ emissions after 2007 reduces the simulated trend in springtime AOD

by 40% from 0.15 $\text{decade}^{-1}$ to 0.09 $\text{decade}^{-1}$ in better agreement with observations (Fig. S3). In contrast, the simulated trend of the springtime aerosol effect changes by less than 15% relative to the simulation driven by CMIP6 emissions. This primarily reflects the decrease of both black carbon and $SO_2$ emissions starting in 2007 (Fig. 1), which results in opposite changes in the aerosol effect. Similar to India (in winter), the discrepancy between the model performances for AOD and $\text{DRE}_{\text{sw}}^{\text{clr}}$ trends points to a bias in aerosol properties. In particular, MEIC suggests that BC emissions have remained stable from 2005 up to 2013. If instead BC emissions increased over this time period as in the historical CMIP6 emissions, the change in the simulated $\text{DRE}_{\text{clr}}^{\text{sw}}$ would be reduced without significant impact on the simulated AOD.

Errors in the representation of the photochemical production of aerosols may also contribute to the model bias. Recent studies have shown that the heterogeneous oxidation of $SO_2$ by $NO_2$ (Cheng et al., 2016) and $O_2$ (Hung and Hoffmann, 2015) at the surface of or in aerosols may be an important source of sulfate in the North China Plains (Wang et al., 2014a; Zheng et al., 2015; Guo et al., 2017; He et al., 2017). To examine the sensitivity of our simulation to this chemistry, we perform an additional simulation using MEIC emissions and the parameterization of the heterogeneous production of sulfate on aerosols from Zheng et al. (2015) (Fig. 8). We find that the heterogeneous oxidation of sulfate increases the simulated sulfate optical depth by 100% in winter and 62% in fall, relative to the simulation driven by MEIC emissions. In contrast, changes are much smaller ($< 25\%$) in spring and summer, which reflects the greater availability of oxidants. The increased production of sulfate in winter and fall results in a stronger link between $SO_2$ emissions and the simulated AOD and aerosol effect (Fig. 8). This stronger link allows the model to better capture some prominent features in the observational record, such as the dip in the aerosol effect in fall 2006 (peak in AOD) or the AOD decrease after 2013. This suggests that both changes to the CMIP6 emissions and to the representation of $SO_2$ photochemistry are needed for AM3 to capture observed changes in the aerosol effect over China from 2001 to 2015.

## 4  Implication for the aerosol direct forcing

In section 3.2, we have shown that regional differences in the speciation of anthropogenic emissions (e.g., the ratio of ammonia and BC to $SO_2$) and the oxidative environment are important to understand changes in the direct shortwave aerosol radiative effect under clear-sky over the largest sources of anthropogenic pollution.

Fig. 9 shows that the changes in the meridional distribution of BC, NO, $NH_3$ and $SO_2$ anthropogenic emissions between 1850 and 2001 (panel a) and between 2001 and 2015 (panel b). In particular, the 2001–2015 period is characterized by higher emissions of BC (25%), NO (15%), and $NH_3$ (19%) and lower $SO_2$ emissions (-12.5%), relative to the 1850–2001 period. While BC and $NH_3$ emissions have increased in most regions, the change in $SO_2$ and NO emissions is associated

with a decline in the northern midlatitudes and an increase in the tropics. Here, we quantify the associated changes in the meridional distribution of the aerosol direct radiative forcing (DRF), the anthropogenic component of the aerosol direct radiative effect.

The aerosol direct radiative forcing for year $y$ is calculated as:

$$DRF(y) = DRE(anthro = y, met = y) - DRE(anthro = 1850, met = y) \qquad (2)$$

where $anthro$ and $met$ denote the year used for anthropogenic emissions and to nudge the horizontal wind, respectively. Note that we use the same meteorology for both simulations, in order to minimize differences in natural sources (e.g., dust, sea salt, dimethylsulfide). On the basis of our evaluation of AM3, we include MEIC emissions for China, the seasonality of $NH_3$ from AIRS in India, and the heterogeneous oxidation of $SO_2$ on aerosol surfaces. We estimate the forcing from biomass burning and non-biomass burning sources separately, as the contribution of anthropogenic activities to changes in biomass burning emissions remains uncertain (Heald et al., 2014). The average 2001–2015 simulated direct radiative forcing from fires is -0.011 $W\,m^{-2}$, which falls within the range of previous model assessments ($0.0 \pm 0.05 W\,m^{-2}$, (Myhre et al., 2013)) . In the following we focus on the radiative forcing from non-biomass burning sources from 1850 to 2001 and from 2001 to 2015.

### 4.1 Clear-sky aerosol direct radiative forcing

The aerosol clear-sky direct radiative forcing in 2001 relative to 1850 is -0.64 $W\,m^{-2}$, which agrees well with previous assessments (Table S4). This forcing is dominated by changes in sulfate (-0.73 $W\,m^{-2}$), which are partly offset by changes in BC (+0.36 $W\,m^{-2}$). Fig. 9c shows that the meridional distribution of the clear-sky radiative forcings of individual aerosols in 2001 relative to 1850 largely mirror that of their precursors' emissions. Some deviations can be noted however. For instance, the forcing from black carbon is enhanced at high latitudes because of the higher surface albedo (Myhre et al., 2013).

We find little change in the aerosol clear-sky direct radiative forcing in 2015 relative to 2001 (-0.04 $W\,m^{-2}$) consistent with previous studies (Murphy, 2013; Kühn et al., 2014). In AM3, this reflects the cancellation between the positive clear-sky aerosol direct radiative forcing in the northern midlatitudes (associated with the decrease of sulfate and the increase of BC) and the negative clear-sky aerosol direct radiative forcing in the northern tropics (associated with the increase of nitrate and sulfate aerosols).

Next we examine the sensitivity of individual forcings to anthropogenic emissions in both periods. The clear-sky direct radiative forcing of black carbon in 2015 relative to 2001 is 25% of the forcing in 2001 relative to 1850, in good agreement with the change in BC emissions. In contrast, the clear-sky direct radiative forcing of sulfate changes little between 2001 and 2015 (+3%), while $SO_2$ emissions decline by -12.5% over the same time period. This small change in the sulfate forcing reflects the cancellation between opposite changes in the tropics, where the forcing from sulfate aerosols is

negative, and the midlatitudes, where it is positive. AM3 shows a stronger sensitivity of the sulfate forcing to changes in $SO_2$ emissions in the tropics than in the midlatitudes. This difference can be attributed to regional differences in the oxidative environment, as a) greater actinic flux allows for more efficient oxidation of $SO_2$ in the tropics than in the midlatitudes (Fig. S4), and b) the efficiency of the oxidation of $SO_2$ to sulfate tends to increase with decreasing $SO_2$ emissions, as oxidant limitations become less important, which diminishes the sensitivity of sulfate to changes in $SO_2$ emissions in the midlatitudes (Fig. S4).

In contrast to sulfate, the change in the clear-sky direct radiative forcing from nitrate from 2001 to 2015 (+75%) is greater than the change in the emissions of its precursors (ammonia and NO emissions increase by less than 20%). The higher sensitivity of nitrate to emission changes in the 2001-2015 period is consistent with the decrease of sulfate in the northern midlatitudes, which enables more ammonia to react with nitric acid to produce ammonium nitrate (Ansari and Pandis, 1998). In the tropics, ammonia is less limiting (the ratio of ammonia to $SO_2$ emissions is higher) and the magnitude of both nitrate and sulfate forcings are simulated to increase from 2001 to 2015.

### 4.2  All-sky aerosol direct radiative forcing

Clouds can enhance the reflectivity of the surface beneath aerosols as well as mask the effect of aerosols underneath (Heald et al., 2014). Overall, clouds tend to amplify the forcing of absorbing aerosols and diminish that of scattering aerosols. The simulated aerosol forcing in 2001 is -0.09 $\mathrm{W\,m^{-2}}$, at the low end of previous multi-model assessments (-0.27 $\pm 0.15 \mathrm{W\,m^{-2}}$ (Myhre et al., 2013) and Table S4) switching sign from negative to positive North of $45°$. For comparison, the instantaneous radiative forcing from well-mixed greenhouse gases at TOA, as calculated from the GFDL Standalone radiation code (Schwarzkopf and Ramaswamy, 1999; Freidenreich and Ramaswamy, 1999), is +1.84 $\mathrm{W\,m^{-2}}$ in 2001.

From 2001 to 2015, the direct aerosol forcing is simulated to be +0.03 $\mathrm{W\,m^{-2}}$, including +0.12, -0.03, and -0.03 $\mathrm{W\,m^{-2}}$ from black carbon, sulfate, and nitrate, respectively. Myhre et al. (2017) recently reported a similar change in the overall direct radiative forcing (+0.01 $\mathrm{W\,m^{-2}}$) but different contributions from sulfate (+0.03 $\mathrm{W\,m^{-2}}$) and black carbon (+0.03 $\mathrm{W\,m^{-2}}$). Many factors could contribute to these differences including the radiative properties of aerosols (e.g., the mixing of sulfate with black carbon (Bond et al., 2013)) and the emission inventories. Further studies are needed to examine whether changes in the sensitivity of radiative forcing to anthropogenic emissions are robust across models. Such assessment would be especially important in the northern midlatitudes, where the direct radiative forcing from aerosols and greenhouse gases from 2001 to 2015 are simulated to be of similar magnitude (+0.25 $\mathrm{W\,m^{-2}}$).

## 5 Conclusions

We have derived estimates of the changes in the aerosol direct clear-sky shortwave radiative effect from 2001 to 2015 using variations in the outgoing shortwave clear-sky radiation from CERES-EBAF. Even over polluted regions, such changes can not be solely ascribed to aerosols and the impact of changes in surface albedo, water vapor and ozone on outgoing radiation need to be accounted for. In particular, we have shown that the effect of increasing anthropogenic aerosols on the outgoing radiation has been largely masked by a decrease in surface albedo over India.

We have used observed seasonal changes in AOD and aerosol effect over large source regions of anthropogenic emissions to assess the representation of anthropogenic emissions and their impact on atmospheric chemistry and the aerosol direct radiative effect in the GFDL-AM3 global chemistry-climate model. Such observational constraints may be especially valuable for future multi-model assessments.

Our work suggests a mature understanding of changes in the aerosol effect over the US and Europe, where the decrease of sulfate aerosols accounts for most of the increase (i.e., the weakening) in the aerosol direct clear-sky shortwave radiative effect. In contrast, the different mix of anthropogenic emissions in India and China results in a more complex speciation of aerosols responsible for changes in the aerosol direct effect, with large contributions from sulfate, nitrate, and black carbon. Trends in these regions remain challenging to capture in the GFDL AM3 model. Some of these biases may be model-specific, including the treatment of the mixing between sulfate and black carbon or the representation of the photochemistry of sulfate and nitrate. Others are attributed to the CMIP6 emissions and will likely affect other models. In particular, we find that the model bias in winter over India can be largely accounted for by uncertainties in the seasonality of ammonia and black carbon emissions. Similarly, comparisons between the CMIP6 and MEIC emission inventories over China suggest that the model bias in this region can be largely attributed to an underestimate in CMIP6 of the reduction of $SO_2$ emissions after 2007.

Our study shows that regional differences in the emission mix and oxidative conditions have a large impact on the relationship between anthropogenic emissions and direct aerosol forcing. Specifically, we have shown that changes in the magnitude, speciation, and spatial distribution of anthropogenic emissions have dampened the sensitivity of the aerosol forcing to $SO_2$ emissions, but amplified that to emissions of NO and ammonia, the precursors of nitrate aerosols. This suggests that relationships between anthropogenic emissions and aerosol forcing derived over the 1850–2001 period and thus largely controlled by changes of $SO_2$ in Europe and North America (Stevens and Schwartz, 2012) need to be revisited with an emphasis on black carbon and ammonia in Asia.

*Acknowledgements.* We thank the many researchers, who have contributed to the CERES, MODIS, and MISR products used in this study. CERES data were obtained from the NASA Langley Research Center CERES ordering tool at http://ceres.larc.nasa.gov/. MODIS albedo (MD43C3 MODIS/Terra+Aqua BRDF/Albedo Albedo Daily L3 Global 0.05 Deg CMG V006) was obtained in netCDF file format from the Integrated Climate Data Center (ICDC, http://icdc.cen.uni-hamburg.de, University of Hamburg, Hamburg, Germany). MISR and

MODIS AOD products can be obtained from the NASA Earthdata portal. Model outputs are available upon request to Fabien.Paulot@noaa.gov. We thank Drs B. Zheng and Q. Zhang for providing MEIC gridded emissions. This work was supported by NOAA Climate Program Office. P. G. acknowledges partial funding by NASA through NNH14ZDA001N-ACMAP grant. We thanks Dr A. Jones, Dr. L.J. Donner, and two anonymous reviewers for helpful comments. All figures were generated using the NCAR Command Language (Ver-

sion 6.4.0, http://dx.doi.org/10.5065/D6WD3XH5).

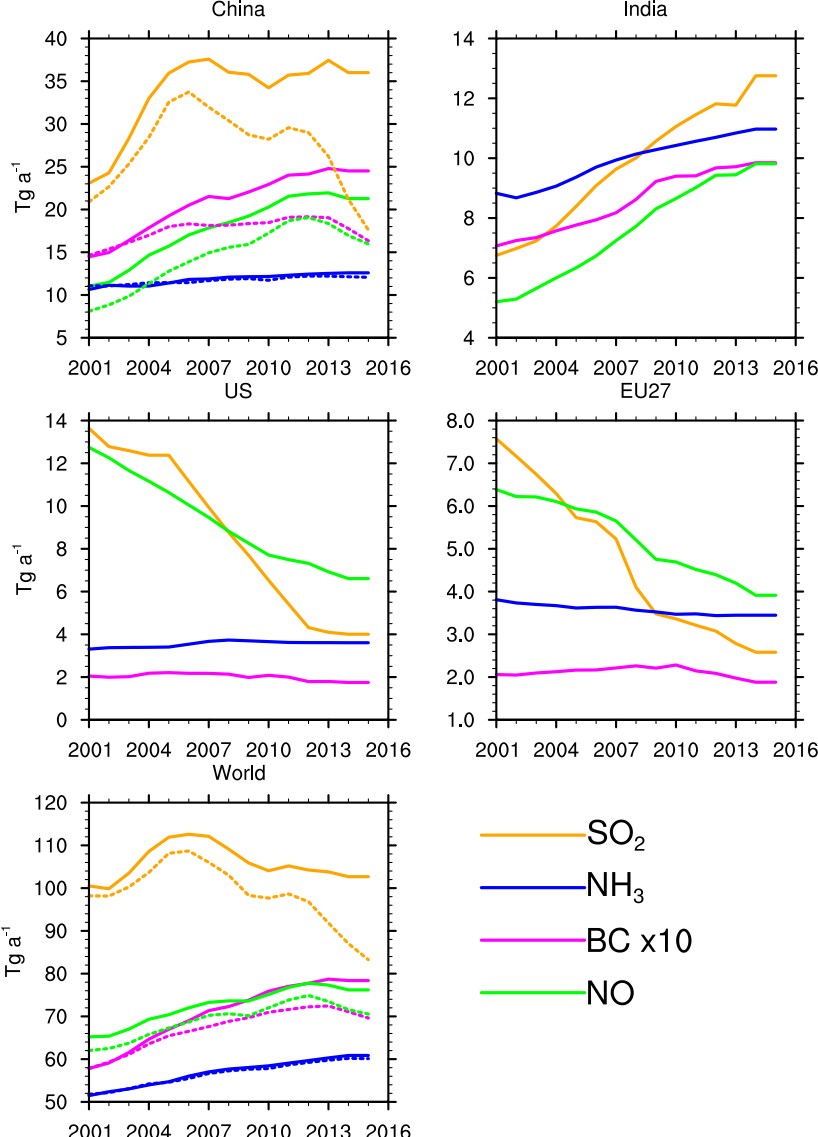

**Figure 1.** Annual anthropogenic emissions of $SO_2$, BC, $NH_3$, and NO from CMIP6 (solid lines) in selected regions. Emissions of $SO_2$, and NO with anthropogenic emissions from MEIC (for agriculture, energy, transportation, industry, and residential sectors) are also shown (dash lines).

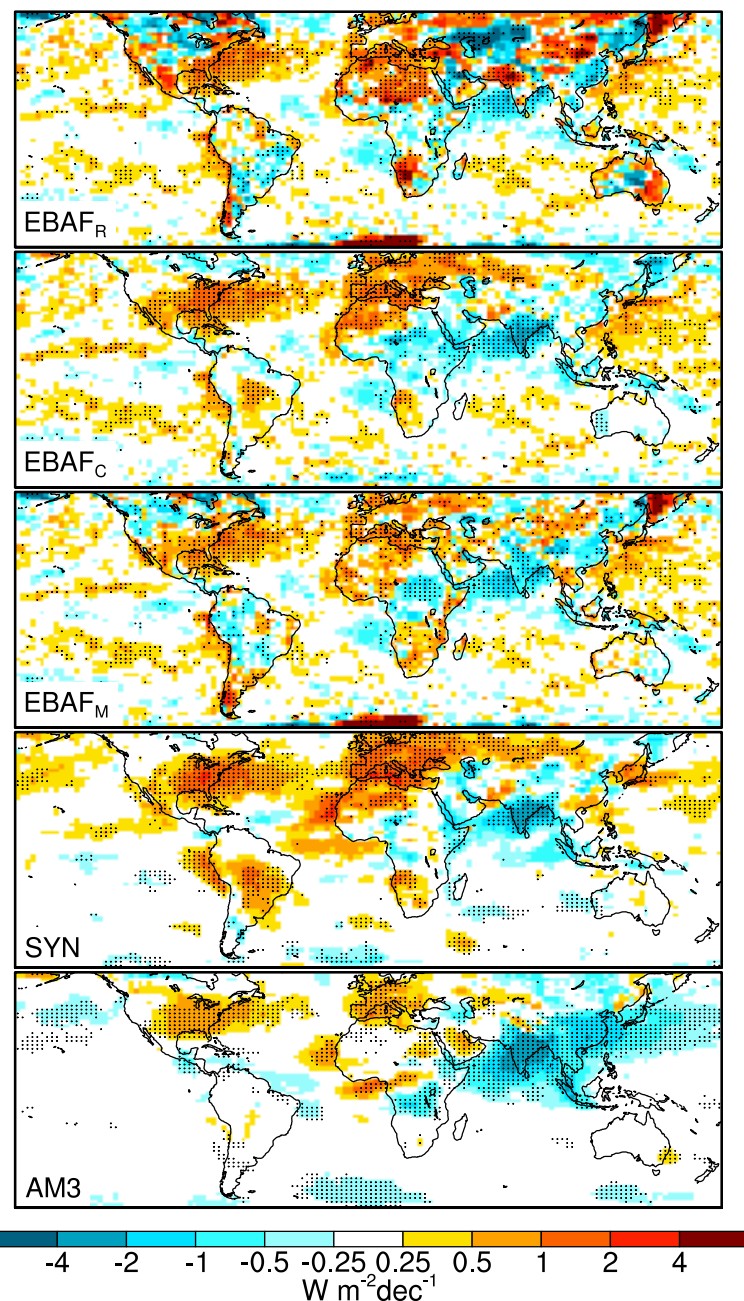

**Figure 2.** Rate of change in the clear-sky shortwave aerosol shortwave direct radiative effect ($\mathrm{DRE^{sw}_{clr}}$). An increase in $\mathrm{DRE^{sw}_{clr}}$ reflects a decrease in the amount of radiation scattered to space by aerosols. $\mathrm{EBAF_R}$ is based on the outgoing clear-sky shortwave radiation from CERES EBAF assuming its variability is solely associated with aerosols. $\mathrm{EBAF_C}$ and $\mathrm{EBAF_M}$ are estimated using the observed clear-sky outgoing shortwave fluxes from CERES EBAF after accounting for the variability of water vapor, ozone, and surface albedo from CERES-EBAF and MODIS, respectively. Estimates from SYN (calculation constrained by observations) and from the GFDL AM3 global chemistry-climate model are also shown. Dotted areas are significant at the 95% confidence level.

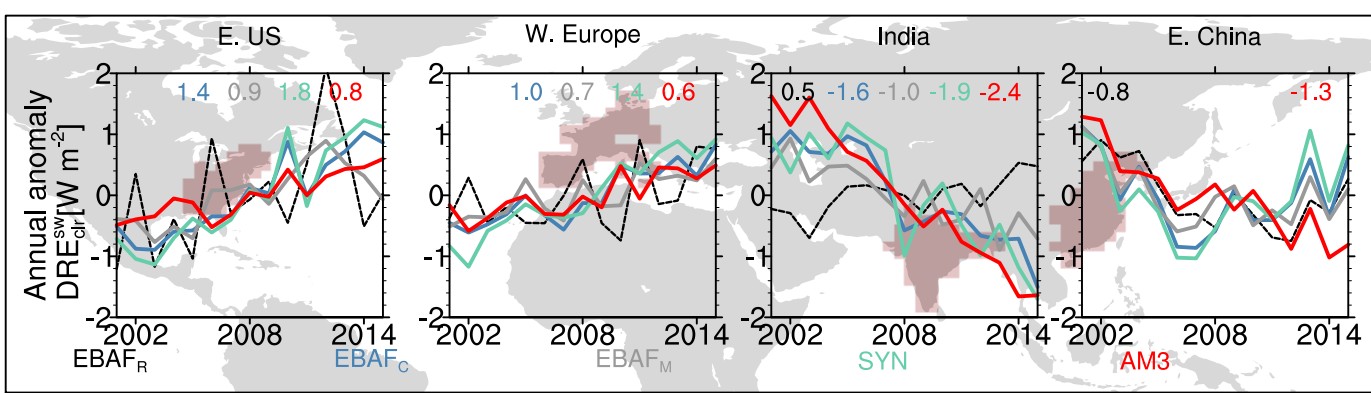

**Figure 3.** Regional changes in the clear-sky shortwave aerosol direct radiative effect derived from CERES-EBAF outgoing radiation without correction (EBAF$_R$ (black)) and after correcting for the variability of water, ozone, and surface albedo (from CERES-EBAF (EBAF$_C$, blue) or from MODIS (EBAF$_M$, grey)) over the Eastern US, Western Europe, India, and Eastern China. Estimates from SYN (calculation constrained by observations) and from the GFDL AM3 global chemistry-climate model are shown in green and red respectively. The rate of change for each estimate is indicated in $\mathrm{W\,m^{-2}\,decade^{-1}}$ when significant ($p$<0.05).

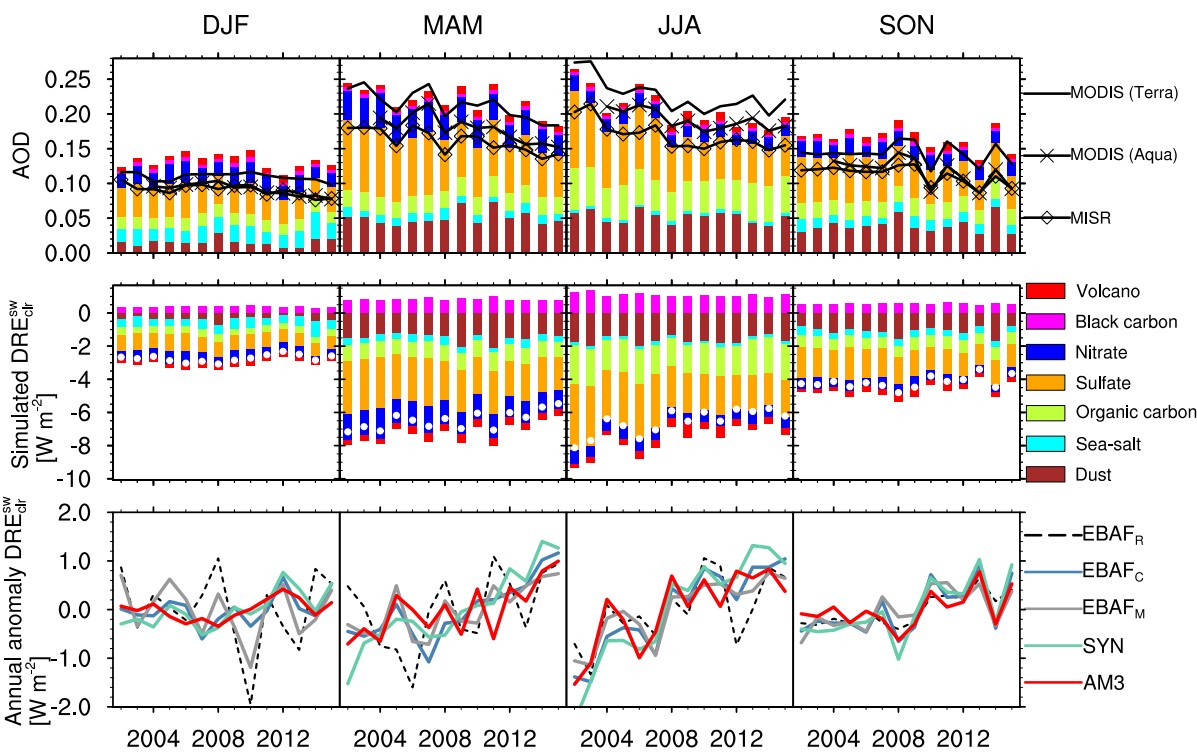

**Figure 4.** Seasonal changes in the aerosol optical depth (AOD) and clear-sky shortwave aerosol direct radiative effect ($DRE_{clr}^{sw}$) in Western Europe (Fig. 3). The top row shows the AOD retrieved from different spaceborne instruments (MODIS-Terra (lines), MODIS-Aqua (cross), MISR (diamond)) and the simulated AOD decomposed into its components (bars). The second row shows the simulated clear-sky shortwave aerosol direct radiative effect of individual aerosols (bars) and the overall aerosol direct radiative effect (white circle). The bottom row shows observation-based and simulated estimates of changes in the aerosol direct radiative effect.

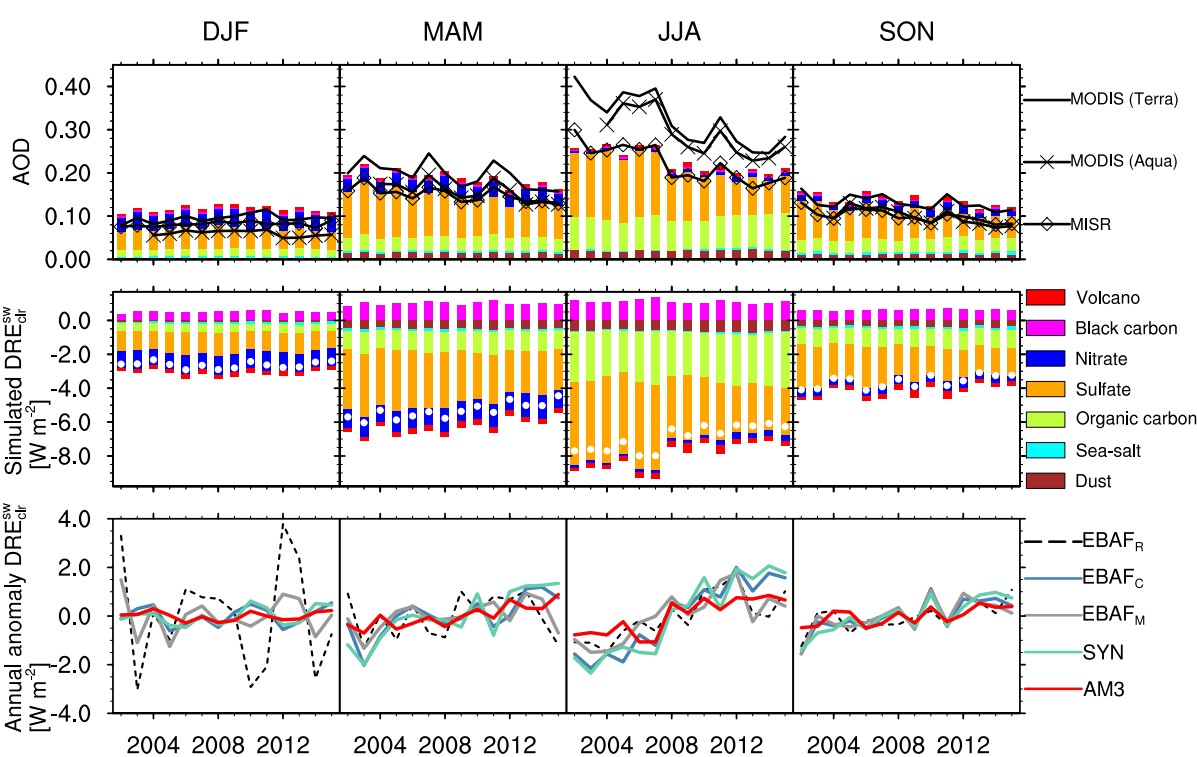

**Figure 5.** Same as 4 for the Eastern US

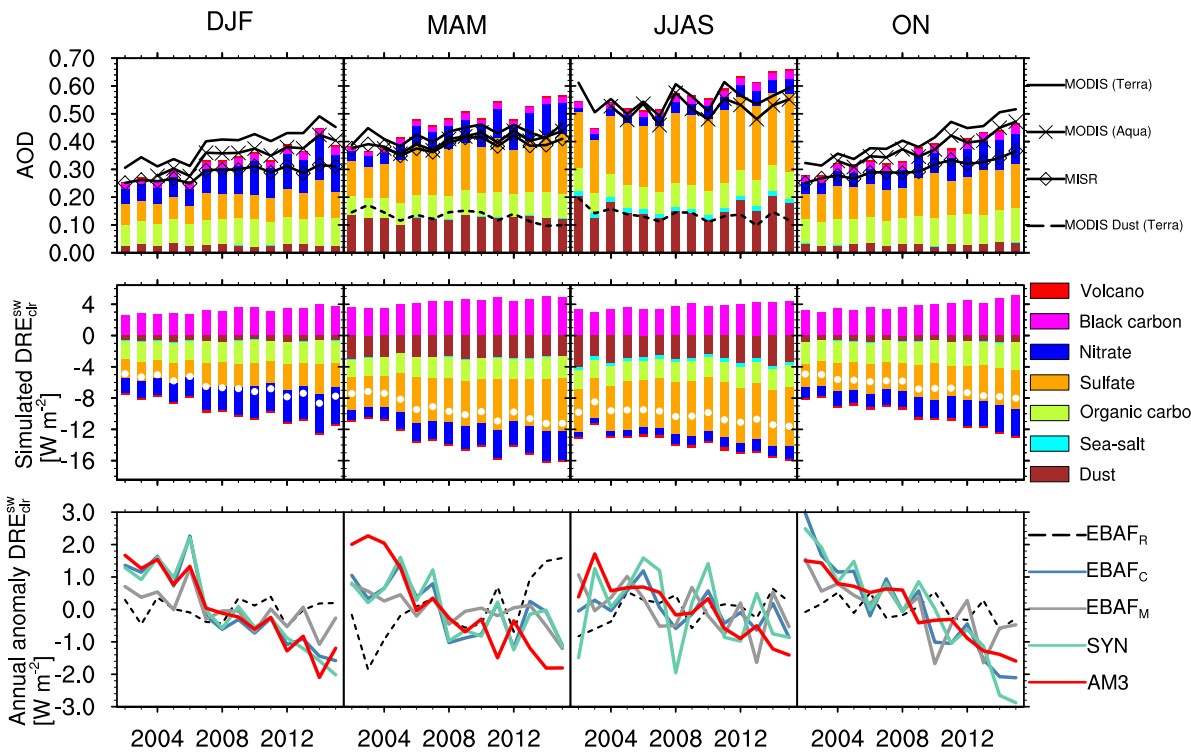

**Figure 6.** Same as 4 for the India. MISR is excluded in the monsoon season, when its coverage is too sparse relative to MODIS (TERRA). The MODIS-derived dust optical depth is indicated by a black dash line.

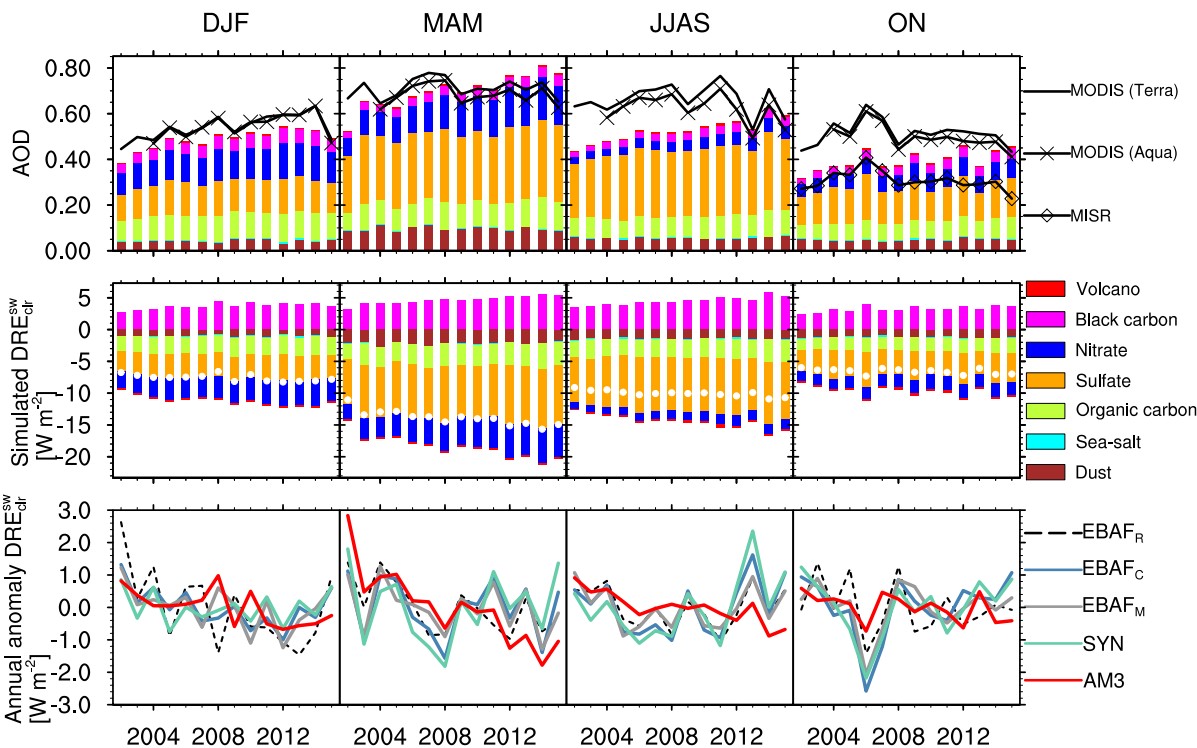

**Figure 7.** Same as 4 for Eastern China. MISR is excluded in winter, spring, and monsoon seasons, when its coverage is too sparse.

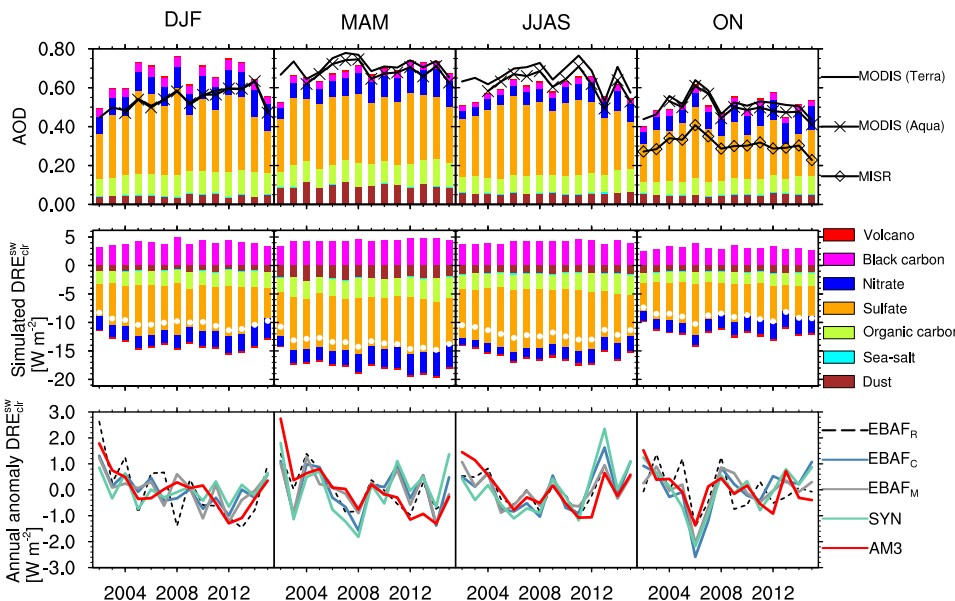

**Figure 8.** Same as Fig. 7 but including the heterogeneous oxidation of $SO_2$ and MEIC emissions over China (see text)

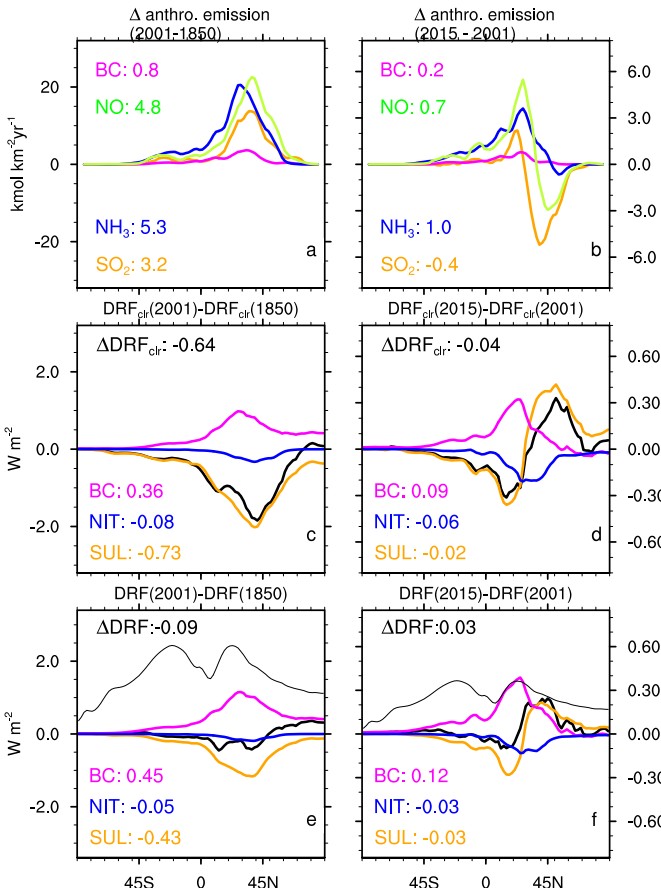

**Figure 9.** Meridional distribution of changes in anthropogenic emissions (BC, NO, $NH_3$, and $SO_2$) and in clear-sky ($DRF_{clr}$, middle row) and all-sky radiative aerosol direct radiative forcing (DRF, bottom row) from 1850 to 2001 (left) and from 2001 to 2015 (right). The thin black line indicates the instantaneous radiative forcing at TOA from well-mixed greenhouse gases. Global anthropogenic emissions and the total and speciated $DRF_{clr}$ and DRF are indicated inline.

**Table 1.** Trends in the aerosol optical depth (AOD, $\mathrm{decade}^{-1}$), and direct clear-sky shortwave radiative effect ($\mathrm{DRE_{clr}^{sw}}$, $\mathrm{W\,m^{-2}\,decade^{-1}}$) for selected regions and seasons from 2002 to 2015[a]

| | Western Europe | | Eastern US | | India | | Eastern China |
| | MAM | JJA | MAM | JJA | DJF | MAM | MAM |
|---|---|---|---|---|---|---|---|
| AOD | | | | | | | |
| MODIS (TERRA) | -0.04 [0.21] | -0.04 [0.23] | -0.04 [0.20] | -0.11 [0.32] | 0.13 [0.39] | 0.04 [0.43] | * [0.71] |
| MODIS (AQUA) | -0.05 [0.18] | -0.03 [0.19] | -0.04 [0.16] | -0.10 [0.29] | 0.11 [0.35] | 0.07 [0.40] | * [0.68] |
| MISR | -0.03 [0.16] | -0.03 [0.17] | -0.02 [0.15] | -0.08 [0.22] | 0.05 [0.29] | * [0.39] | |
| MATCH[b] | -0.06 [0.27] | -0.06 [0.26] | -0.07 [0.29] | -0.11 [0.35] | 0.10 [0.36] | 0.03 [0.49] | * [0.90] |
| AM3 | -0.04 [0.22] | -0.05 [0.21] | -0.03 [0.19] | -0.05 [0.23] | 0.13 [0.33] | 0.15 [0.47] | 0.15 [0.70] |
| sulfate | -0.03 [0.08] | -0.04 [0.07] | -0.03 [0.09] | -0.06 [0.12] | 0.02 [0.09] | 0.07 [0.17] | 0.05 [0.30] |
| nitrate | -0.01 [0.04] | * [0.02] | * [0.03] | 0.00 [0.01] | 0.07 [0.10] | 0.06 [0.07] | 0.08 [0.14] |
| black carbon | * [0.01] | * [0.00] | * [0.01] | * [0.00] | 0.01 [0.02] | 0.01 [0.02] | 0.01 [0.04] |
| $\mathrm{DRE_{clr}^{sw}}$ | | | | | | | |
| SYN | 1.8 [-8.9] | 2.5 [-9.4] | 2.1 [-8.6] | 3.6 [-11.0] | -2.6 [-9.1] | -1.4 [-13.4] | * [-20.5] |
| $\mathrm{EBAF_C}$ | 1.4 | 1.8 | 1.3 | 3.3 | -2.3 | -1.2 | * |
| $\mathrm{EBAF_M}$ | 1.0 | 1.2 | * | 2.0 | -0.8 | -0.9 | * |
| AM3 | 1.1 [-6.5] | 1.5 [-6.6] | 0.9 [-5.3] | 1.4 [-6.9] | -2.7 [-6.6] | -3.1 [-9.4] | -2.1 [-13.9] |
| sulfate | 0.9 [-2.6] | 1.5 [-2.7] | 1.1 [-3.1] | 2.2 [-3.9] | -0.7 [-2.9] | -1.8 [-5.5] | -1.1 [-8.5] |
| nitrate | 0.3 [-1.4] | * [-0.7] | * [-1.2] | -0.2 [-0.2] | -2.4 [-3.3] | -1.9 [-2.6] | -2.2 [-4.2] |
| black carbon | * [0.8] | -0.2 [1.1] | * [1.0] | * [1.1] | 0.9 [3.2] | 1.2 [4.3] | 1.4 [4.7] |

[a] The average over the period 2002–2015 is shown in bracket (2003-2015 for AQUA). Trends are estimated using the Theil-Sen method. * denotes non significant monotonous change at $p = 0.05$. Model AOD is sampled based on MODIS (TERRA) seasonal coverage. No statistics is provided for China from MISR because of large differences in spatial coverage with MODIS (TERRA). SYN refers to the aerosol effect calculated in the CERES-SYN product. $\mathrm{EBAF_C}$ and $\mathrm{EBAF_M}$ refer to the aerosol effect estimated using CERES-EBAF outgoing shortwave clear-sky radiation corrected for the variability in water, ozone, and CERES-EBAF ($\mathrm{EBAF_C}$) and MODIS ($\mathrm{EBAF_M}$) surface albedo. Confidence intervals for the trends are provided in Table S2.

[b] from CERES-SYN Ed4 based on assimilation of MODIS Collection5 AOD with the MATCH model.

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
