# Peer review of "Changes in the aerosol direct radiative forcing from 2001 to 2015: observational constraints and regional mechanisms"

_Atmospheric Chemistry and Physics, 2018_

## Referee Comment (RC1) · Anonymous Referee #1 · 10 Mar 2018

This paper discusses the use of the GFDL-AM3 model for interpreting changes in the outgoing radiation budget as a result of changes in aerosol emissions. The direction of the paper is very interesting as a result as it could provide a basis for updating the direct radiative forcing estimates from aerosols. However, at this time this paper needs substantial revisions before it can be accepted for publication or even given a serious review. In particular there are two main concerns that I have. Concern 1 is that the paper is nearly un-readable with the extensive use of acronyms and jargon, which in turn makes it challenging to understand their conclusions. Concern 2 is that the paper effectively does not consider the role of fires in their model / data comparisons. Consequently, the comparisons between data and model over India Asia (in particular)

and anywhere in the tropics will be substantively compromised (e.g. Ramanathan and Carmichael, Nat Geo 2008 and references therein). In fact it seems likely that their poor model/data comparison in the tropics is likely because fires are not obviously considered in the comparison.

Below are examples of sentences and paragraphs that need to be made more clear. I should point out that their writing approach is infused in nearly every section so the authors really just need to examine every single sentence in this paper and determine how to "de-acronym" and "de-jargon" the sentence.

I think this sentence is possibly the worst offender: Line 155: "Figure 2 shows that $\Delta m,yRstucsaf$ is most sensitive to $\Delta m,y(salb)$, so we use two independent estimates of surface albedo derived from CERES EBAF (salbCE) and MODIS (salbM). The associated estimates of SDRECS will be denoted as SDRECSCE and SDRECSM, respectively, hereafter. We use GEOS5 for WVP and qo3, similar to CERES SYN. "

Below are other "minor" comments. I should point out that I can probably cut out a large fraction of the paper and attempt to make suggestions about changing the "readability" and grammar. However, I would really want to see this paper cleaned up and my concern about fires addressed before going through and providing a more thorough review

Line 165: What does this mean? this is apparently one of the many "p-tests" that gets used about the literature but essentially has no meaning if not explained.

Line 216: (fix this statement) "Changes in AOD are dominated by spring and summer" Line 231: ??? Again too much acronyms / jargon. What is conclusion about these differences? Line 353. . . no biomass burning as part of anthropogenic emissions? The bulk of fires is due to human activities so this is an odd statement. Perhaps you mean the subset that is not human driven? That said, you would see substantial differences between observed and actual outgoing radiation if you leave this term out (which you apparently do over the tropics).

---

## Referee Comment (RC2) · Anonymous Referee #2 · 16 Apr 2018

This paper presents an important contribution to the effort to understand current trends in aerosol loadings and their associated radiative forcing. The authors use a combination of recent observational datasets, the CEDS emission inventory and the atmospheric component of the GFDL model. They find overall consistency in regional trends between the methods, but also identify potential issues with the emission inventory. They are also able to subdivide the trends into contributions from individual aerosol components, so far as the accuracy of a single model can go.

Unfortunately, it is at present also a very difficult paper to read and understand. Even after multiple readings, there are several details I struggle to understand, and some

conclusions that I can't quite grasp how the authors have arrived at. Hence, while I realize it will require some work, I strongly recommend that the authors fully revise the paper for clarity and readability. This will make the important results presented here much more accessible to the community.

Some examples and comments:

- A good illustration is the term SDRECS. While it is well defined in the paper, it is technical and not standard in similar literature. Why not simply write "change in outgoing radiation"? Most of the paper deals with shortwave under clear sky conditions, so this is implicit even from the title. The same goes for Rsutcsaf , Rsutcs and similar.

- One challenge, especially in the latter part of the paper (the regional trends and RF discussions), is to follow where the conclusions depend on the specific aerosol parametrizations of the GFDL model, and where they can be assumed to be more general. I would encourage the authors to add some further discussion of how model dependent the conclusions are. E.g. in the Conclusions, how general are the remarks about possible issues with the CEDS inventory? This is an important discussion for a dataset that will form the basis for much of CMIP6. A specific example: The authors conclude that "we find significant uncertainties in the CMIP6 emissions, including in the seasonality of NH3". In the paper, as far as I can understand, this is documented through the following: "We conducted a sensitivity simulation using the seasonality of NH3 column from AIRS (Warner et al., 2017) 265 to modulate NH3 emissions. We find that this revised seasonality significantly reduces the simulated winter trend in SDRECS (0.08 Wm−2 dec−1 ), improving the agreement with observations." I would expect some more discussion and documentation on this point, to make such a broad conclusion.

- A more technical example: In Figure 1, introduced on line 177, the authors show both CEDS and MEIC emissions. However, "MEIC" isn't defined or discussed until line 300, making it difficult to understand even the first figure without already having read the

entire manuscript. Please review for clarity, with a community reader in mind.

- For the DRF discussion, it would be good to put the results in a broader context. AeroCom is mentioned; where is AM3 relative to the model mean in terms of forcing strengths? E.g. a comparison to the similar (but much less detailed) results in Myhre et al 2017, ACP ( https://www.atmos-chem-phys.net/17/2709/2017/) would be useful.

In conclusion, I hope the authors take the time to revise the paper for clarity. As it stands, it will be difficult for the community to access - which is a shame, as the results are important and of broad interest.

---

## Author Comment (AC1) · 21 May 2018

We thank both reviewers for their comments.
Replies and changes are listed below.

**1    General presentation**

Following both reviewers' comments, the manuscript has been revised extensively. Most significantly,

1. The usage of acronyms has been greatly reduced in the text, in order to improve readability. In figures, we have chosen to replace SDRECS by the more compact $DRE_{clr}^{sw}$.

2. The conclusions have been revised to better emphasize, the results that may be model specific from those that are applicable to others models participating in CMIP6.

**2    Reviewer 1**

1. **the paper effectively does not consider the role of fires in their model / data comparisons. Consequently, the comparisons between data and model over India Asia (in particular) and anywhere in the tropics will be substantively compromised (e.g. Ramanathan and Carmichael, Nat Geo 2008 and references therein). In fact it seems likely that their poor model/data comparison in the tropics is likely because fires are not obviously considered in the comparison.**
Our model includes monthly biomass burning emissions from Global Fire Emissions Database. We agree with the reviewer that uncertainties in biomass burning could contribute to model biases in tropical regions. However, decadal changes in biomass burning are small relative to those in anthropogenic emissions in the regions that we focus on (India, China, the Eastern US, and Europe), which suggests it is unlikely to contribute to errors in the simulated trends in the aerosol effect. We have have added a reference to the van der Werf et al. [2017], which describes the methodology used to derived GFED emissions and highlighted that it is based on satellite observations. See also reply to comment 5 and to comment 4 from reviewer 2.
The text was revised as follow:

*Monthly biomass burning emissions are from the historical global biomass burning emissions inventory for CMIP6 (BB4CMIP6, van Marle et al. [2017]). Emissions for the 1997 to 2015 period in this inventory have been derived from satellite-based emissions from the Global Fire Emissions Database (GFED, van der Werf et al. [2017]).*

2. **Line 165: What does this mean? this is apparently one of the many "p-tests" that gets used about the literature but essentially has no meaning if not explained.**
We have revised the text as follow:

*We use the non-parametric Mann-Kendall test [Kendall, 1938] to identify significant changes in the aerosol effect. This test quantifies monotonic correlations between two variables. It is based on a rank procedure that makes it less susceptible to outliers than the Pearson correlation and thus especially well-suited for the analysis of environmental dataset. Here, we use a critical p value of 0.05 for trend significance. When a significant trend is detected, we estimate the linear trend using the Theil-Sen method [Theil, 1950, Sen, 1968].*

3. **Line 216: (fix this statement) "Changes in AOD are dominated by spring and summer"**
We have revised the text as follow:

*Observations show that the AOD decreases most in spring and summer*

4. **Line 231: ??? Again too much acronyms / jargon. What is conclusion about these differences?**
We have revised the text as follow:

*In both Europe and the US, we find that the change in the aerosol effect inferred from the SYN calculation is larger than that estimated from CERES-EBAF outgoing radiation corrected for surface albedo changes ($EBAF_C$ and $EBAF_M$). The magnitude of the changes in the MATCH AOD, which is used to calculate the SYN estimate, is also greater than in more recent retrieval of AOD from MODIS (Table 1). This suggests that the rate of change in SYN aerosol effect may be biased high in Europe and Western Europe.*

5. **Line 353. . . no biomass burning as part of anthropogenic emissions? The bulk of fires is due to human activities so this is an odd statement. Perhaps you mean the subset that is not human driven? That said, you would see substantial differences between observed and actual outgoing radiation if you leave this term out (which you apparently do over the tropics).**
We have performed an additional simulation to estimate the forcing from biomass burning. The text was revised as follow:

*We estimate the forcing from biomass burning and non-biomass burning sources separately, as the contribution of anthropogenic activities to changes in biomass burning emissions remains uncertain [Heald et al., 2014]. The average 2001–2015 simulated direct radiative forcing from fires is -0.011 $\mathrm{W\,m}^{-2}$, which falls within the range of previous model assessments ($0.0 \pm 0.05\mathrm{W\,m}^{-2}$, [Myhre et al., 2013]).*

**2.1 Reviewer 2**

1. **Readability: A good illustration is the term SDRECS. While it is well defined in the paper, it is technical and not standard in similar literature. Why not simply write "change in outgoing radiation"? Most of the paper deals with shortwave under clear sky conditions, so this is implicit even from the title. The same goes for Rsutcsaf, Rsutcs and similar.**
The manuscript has been modified significantly to improve readability. Following both reviewers' recommendations most acronyms from the main text have been removed. There is no widely accepted acronym to designate the clear-sky shortwave direct aerosol effect. In figures, we have replaced SDRECS by $\mathrm{DRE}_{\mathrm{clr}}^{\mathrm{sw}}$, which we think is easier to understand. We have kept rsutcsaf and rsutcs notations as they are based on *CMIP6* naming convention (http://clipc-services.ceda.ac.uk/dreq/index/CMORvar.html). This is now clearly stated.

2. **One challenge, especially in the latter part of the paper (the regional trends and RF discussions), is to follow where the conclusions depend on the specific aerosol parametrizations of the GFDL model, and where they can be assumed to be more general. I would encourage the authors to add some further discussion of how model dependent the conclusions are. E.g. in the Conclusions, how general are the remarks about possible issues with the CEDS inventory? This is an important discussion for a dataset that will form the basis for much of CMIP6. A specific example: The authors conclude that "we find significant uncertainties in the CMIP6 emissions, including in the seasonality of NH3". In the paper, as far as I can understand, this is documented through the following: "We conducted a sensitivity simulation using the seasonality of NH3 column from AIRS (Warner et al., 2017) 265 to modulate NH3 emissions. We find that this revised seasonality significantly reduces the simulated winter trend in SDRECS (0.08 Wm − 2 dec − 1 ), improving the agreement with observations." I would expect some more discussion and documentation on this point, to make such a broad conclusion.**

   (a) We have added a figure in the supplementary materials comparing the seasonality of ammonia emissions modulated using CMIP6 seasonality and AIRS seasonality (see Fig. 1 below).

   (b) We have revised the conclusion to emphasize that some of the biases may be model specific while others are associated with CMIP6 emissions and will thus affect all models. The text was revised as follow:

      *Some of these biases may be model-specific, including the treatment of the mixing between sulfate and black carbon or the representation of the photochemistry of sulfate and nitrate. Others are attributed to the CMIP6 emissions and will likely affect other models. In particular, we find that the model bias in winter over India can be largely accounted for by uncertainties in the seasonality of ammonia and black carbon emissions. Similarly, comparisons between the CMIP6 and MEIC emission inventories over China suggest that the model bias in this region can be largely attributed to an underestimate of the reduction of $\mathrm{SO}_2$ emissions after 2007.*

3. **For the DRF discussion, it would be good to put the results in a broader context. Aero-Com is mentioned; where is AM3 relative to the model mean in terms of forcing strengths? E.g. a comparison to the similar (but much less detailed) results in Myhre et al 2017, ACP ( https://www.atmos-chem-phys.net/17/2709/2017/) would be useful.**
We have added a comparison with the results from Myhre et al. [2017] in the RF section:

[Figure]

Figure 1: CMIP6 ammonia emissions for India with seasonality from CMIP6 (black) and AIRS (red)

*From 2001 to 2015, the direct aerosol forcing is simulated to be $+0.03\,\mathrm{W\,m^{-2}}$, including $+0.12$, $-0.03$, and $-0.03\,\mathrm{W\,m^{-2}}$ from black carbon, sulfate, and nitrate, respectively. Myhre et al. [2017] recently reported a similar change in the overall direct radiative forcing ($+0.01\,\mathrm{W\,m^{-2}}$) but different contributions from sulfate ($+0.03\,\mathrm{W\,m^{-2}}$) and black carbon ($+0.03\,\mathrm{W\,m^{-2}}$). Many factors could contribute to these differences including the radiative properties of aerosols (e.g., the mixing of sulfate with black carbon [Bond et al., 2013]) and the emission inventories. Further studies are needed to examine whether changes in the sensitivity of radiative forcing to anthropogenic emissions are robust across models. Such assessment would be especially important in the northern midlatitudes, where the direct radiative forcing from aerosols and greenhouse gases from 2001 to 2015 are simulated to be of similar magnitude ($+0.25\,\mathrm{W\,m^{-2}}$).*

4. **A more technical example: In Figure 1, introduced on line 177, the authors show both CEDS and MEIC emissions. However, "MEIC" isn't defined or discussed until line 300, making it difficult to understand even the first figure without already having read the entire manuscript. Please review for clarity, with a community reader in mind**
We have added the following text in the method section
*Anthropogenic emissions in India and China are expecte d to be more uncertain than in the US and Europe [Saikawa et al., 2017a,b]. Fig. 1 shows that the regional Modular Emission Inventory for China (MEIC) [Zhang et al., 2009], shows a decline of $SO_2$ emissions starting in 2006 and accelerating in 2012, a decrease of NO after 2012, and near-stable BC emissions after 2007. In 2014, MEIC NO, $SO_2$, and BC emissions are 24%, 48%, and 32% lower than CMIP6 emissions, respectively. $NH_3$ emissions are similar in magnitude but exhibit different seasonality: CMIP6 $NH_3$ emissions peak in spring, while MEIC exhibits a broad peak in summer, consistent with top-down constraints [Paulot et al., 2014, Zhang et al., 2017].*

**References**

T. C. Bond, S. J. Doherty, D. W. Fahey, P. M. Forster, T. Berntsen, B. J. DeAngelo, M. G. Flanner, S. Ghan, B. Kärcher, D. Koch, S. Kinne, Y. Kondo, P. K. Quinn, M. C. Sarofim, M. G. Schultz, M. Schulz, C. Venkataraman, H. Zhang, S. Zhang, N. Bellouin, S. K. Guttikunda, P. K. Hopke, M. Z. Jacobson, J. W. Kaiser, Z. Klimont, U. Lohmann, J. P. Schwarz, D. Shindell, T. Storelvmo, S. G. Warren, and C. S. Zender. Bounding the role of black carbon in the climate system: A scientific assessment. *Journal of Geophysical Research: Atmospheres*, 118(11): 5380–5552, 2013. ISSN 2169-8996. doi: 10.1002/jgrd.50171. URL http://dx.doi.org/10.1002/jgrd.50171.

C. L. Heald, D. A. Ridley, J. H. Kroll, S. R. H. Barrett, K. E. Cady-Pereira, M. J. Alvarado, and C. D. Holmes. Contrasting the direct radiative effect and direct radiative forcing of aerosols. *Atmos.*

*Chem. Phys.*, 14(11):5513–5527, June 2014. ISSN 1680-7324. doi: 10.5194/acp-14-5513-2014. URL http://www.atmos-chem-phys.net/14/5513/2014/.

M. G. Kendall. A new measure of rank correlation. *Biometrika*, 30(1-2):81–93, 1938. doi: 10.1093/biomet/30.1-2.81. URL + http://dx.doi.org/10.1093/biomet/30.1-2.81.

G. Myhre, B. H. Samset, M. Schulz, Y. Balkanski, S. Bauer, T. K. Berntsen, H. Bian, N. Bellouin, M. Chin, T. Diehl, R. C. Easter, J. Feichter, S. J. Ghan, D. Hauglustaine, T. Iversen, S. Kinne, A. Kirkevåg, J.-F. Lamarque, G. Lin, X. Liu, M. T. Lund, G. Luo, X. Ma, T. van Noije, J. E. Penner, P. J. Rasch, A. Ruiz, Ø. Seland, R. B. Skeie, P. Stier, T. Takemura, K. Tsigaridis, P. Wang, Z. Wang, L. Xu, H. Yu, F. Yu, J.-H. Yoon, K. Zhang, H. Zhang, and C. Zhou. Radiative forcing of the direct aerosol effect from AeroCom phase II simulations. *Atmos. Chem. Phys.*, 13(4):1853–1877, February 2013. ISSN 1680-7324.

G. Myhre, W. Aas, R. Cherian, W. Collins, G. Faluvegi, M. Flanner, P. Forster, Ø. Hodnebrog, Z. Klimont, M. T. Lund, J. Mülmenstädt, C. Lund Myhre, D. Olivié, M. Prather, J. Quaas, B. H. Samset, J. L. Schnell, M. Schulz, D. Shindell, R. B. Skeie, T. Takemura, and S. Tsyro. Multi-model simulations of aerosol and ozone radiative forcing due to anthropogenic emission changes during the period 1990–2015. *Atmospheric Chemistry and Physics*, 17(4): 2709–2720, 2017. doi: 10.5194/acp-17-2709-2017. URL https://www.atmos-chem-phys.net/17/2709/2017/.

F. Paulot, D. J. Jacob, R. W. Pinder, J. O. Bash, K. Travis, and D. K. Henze. Ammonia emissions in the United States, European Union, and China derived by high-resolution inversion of ammonium wet deposition data: Interpretation with a new agricultural emissions inventory (MASAGE_NH3). *J. Geophys. Res. Atmos.*, 119(7): 4343–4364, April 2014. ISSN 2169-8996.

Eri Saikawa, Hankyul Kim, Min Zhong, Alexander Avramov, Yu Zhao, Greet Janssens-Maenhout, Jun ichi Kurokawa, Zbigniew Klimont, Fabian Wagner, Vaishali Naik, Larry W. Horowitz, and Qiang Zhang. Comparison of emissions inventories of anthropogenic air pollutants and greenhouse gases in china. *Atmospheric Chemistry and Physics*, 17(10):6393–6421, may 2017a. doi: 10.5194/acp-17-6393-2017. URL https://doi.org/10.5194/acp-17-6393-2017.

Eri Saikawa, Marcus Trail, Min Zhong, Qianru Wu, Cindy L Young, Greet Janssens-Maenhout, Zbigniew Klimont, Fabian Wagner, Jun ichi Kurokawa, Ajay Singh Nagpure, and Bhola Ram Gurjar. Uncertainties in emissions estimates of greenhouse gases and air pollutants in india and their impacts on regional air quality. *Environmental Research Letters*, 12(6):065002, may 2017b. doi: 10.1088/1748-9326/aa6cb4. URL https://doi.org/10.1088/1748-9326/aa6cb4.

Pranab Kumar Sen. Estimates of the Regression Coefficient Based on Kendall's Tau. *J. Am. Stat. Assoc.*, 63(324):1379–1389, December 1968. ISSN 0162-1459. doi: 10.1080/01621459.1968.10480934. URL http://www.tandfonline.com/doi/abs/10.1080/01621459.1968.10480934.

H. Theil. A rank-invariant method of linear and polynomial regression analysis. I. *Nederl. Akad. Wetensch., Proc.*, 53:386–392 = Indagationes Math. 12, 85–91 (1950), 1950. URL http://www.ams.org/mathscinet-getitem?mr=0036489.

G. R. van der Werf, J. T. Randerson, L. Giglio, T. T. van Leeuwen, Y. Chen, B. M. Roğers, M. Mu, M. J. E. van Marle, D. C. Morton, G. J. Collatz, R. J. Yokelson, and P. S. K̃asibhatla. Global fire emissions estimates during 1997–2016. *Earth System Science Data*, 9(2):697–720, 2017. doi: 10.5194/essd-9-697-2017. URL https://www.earth-syst-sci-data.net/9/697/2017/.

Margreet J. E. van Marle, Silvia Kloster, Brian I. Magi, Jennifer R. Marlon, Anne-Laure Daniau, Robert D. Field, Almut Arneth, Matthew Forrest, Stijn Hantson, Natalie M. Kehrwald, Wolfgang Knorr, Gitta Lasslop, Fang Li, Stéphane Mangeon, Chao Yue, Johannes W. Kaiser, and Guido R. van der Werf. Historic global biomass burning emissions for CMIP6 (BB4cmip) based on merging satellite observations with proxies and fire models (1750–2015). *Geosci. Model Dev.*, 10(9):3329–3357, sep 2017. doi: 10.5194/gmd-10-3329-2017. URL https://doi.org/10.5194/gmd-10-3329-2017.

L. Zhang, Y. Chen, Y. Zhao, D. K. Henze, L. Zhu, Y. Song, F. Paulot, X. Liu, Y. Pan, and B. Huang. Agricultural ammonia emissions in China: reconciling bottom-up and top-down estimates. *Atmos. Chem. Phys. Discuss.*, 2017:1–36, August 2017. ISSN 1680-7375. doi: 10.5194/acp-2017-749. URL https://www.atmos-chem-phys-discuss.net/acp-2017-749/.

Q. Zhang, D. G. Streets, G. R. Carmichael, K. B. He, H. Huo, A. Kannari, Z. Klimont, I. S. Park, S. Reddy, J. S. Fu, D. Chen, L. Duan, Y. Lei, L. T. Wang, and Z. L. Yao. Asian emissions in 2006 for the NASA INTEX-B mission. *Atmos. Chem. Phys.*, 9(14):5131–5153, July 2009. ISSN 1680-7324. doi: 10.5194/acp-9-5131-2009. URL https://www.atmos-chem-phys.net/9/5131/2009/.

---

## Author Response (AR1)

We thank both reviewers for their comments.
Replies and changes are listed below.

**1 General presentation**

Following both reviewers' comments, the manuscript has been revised extensively. Most significantly,

1. The usage of acronyms has been greatly reduced in the text, in order to improve readability. In figures, we have chosen to replace SDRECS by the more compact $DRE_{clr}^{sw}$.

2. The conclusions have been revised to better emphasize, the results that may be model specific from those that are applicable to others models participating in CMIP6.

**2 Reviewer 1**

1. **the paper effectively does not consider the role of fires in their model / data comparisons. Consequently, the comparisons between data and model over India Asia (in particular) and anywhere in the tropics will be substantively compromised (e.g. Ramanathan and Carmichael, Nat Geo 2008 and references therein). In fact it seems likely that their poor model/data comparison in the tropics is likely because fires are not obviously considered in the comparison.**
Our model includes monthly biomass burning emissions from Global Fire Emissions Database. We agree with the reviewer that uncertainties in biomass burning could contribute to model biases in tropical regions. However, decadal changes in biomass burning are small relative to those in anthropogenic emissions in the regions that we focus on (India, China, the Eastern US, and Europe), which suggests it is unlikely to contribute to errors in the simulated trends in the aerosol effect. We have have added a reference to the van der Werf et al. [2017], which describes the methodology used to derived GFED emissions and highlighted that it is based on satellite observations. See also reply to comment 5 and to comment 4 from reviewer 2.
The text was revised as follow:

   *Monthly biomass burning emissions are from the historical global biomass burning emissions inventory for CMIP6 (BB4CMIP6, van Marle et al. [2017]). Emissions for the 1997 to 2015 period in this inventory have been derived from satellite-based emissions from the Global Fire Emissions Database (GFED, van der Werf et al. [2017]).*

2. **Line 165: What does this mean? this is apparently one of the many "p-tests" that gets used about the literature but essentially has no meaning if not explained.**
We have revised the text as follow:

   *We use the non-parametric Mann-Kendall test [Kendall, 1938] to identify significant changes in the aerosol effect. This test quantifies monotonic correlations between two variables. It is based on a rank procedure that makes it less susceptible to outliers than the Pearson correlation and thus especially well-suited for the analysis of environmental dataset. Here, we use a critical p value of 0.05 for trend significance. When a significant trend is detected, we estimate the linear trend using the Theil-Sen method [Theil, 1950, Sen, 1968].*

3. **Line 216: (fix this statement) "Changes in AOD are dominated by spring and summer"**
We have revised the text as follow:

   *Observations show that the AOD decreases most in spring and summer*

4. **Line 231: ??? Again too much acronyms / jargon. What is conclusion about these differences?**
We have revised the text as follow:

   *In both Europe and the US, we find that the change in the aerosol effect inferred from the SYN calculation is larger than that estimated from CERES-EBAF outgoing radiation corrected for surface albedo changes ($EBAF_C$ and $EBAF_M$). The magnitude of the changes in the MATCH AOD, which is used to calculate the SYN estimate, is also greater than in more recent retrieval of AOD from MODIS (Table 1). This suggests that the rate of change in SYN aerosol effect may be biased high in Europe and Western Europe.*

5. **Line 353. . . no biomass burning as part of anthropogenic emissions? The bulk of fires is due to human activities so this is an odd statement. Perhaps you mean the subset that is not human driven? That said, you would see substantial differences between observed and actual outgoing radiation if you leave this term out (which you apparently do over the tropics).**
We have performed an additional simulation to estimate the forcing from biomass burning. The text was revised as follow:

*We estimate the forcing from biomass burning and non-biomass burning sources separately, as the contribution of anthropogenic activities to changes in biomass burning emissions remains uncertain [Heald et al., 2014]. The average 2001–2015 simulated direct radiative forcing from fires is -0.011 $\mathrm{W\,m^{-2}}$, which falls within the range of previous model assessments ($0.0 \pm 0.05\mathrm{W\,m^{-2}}$, [Myhre et al., 2013]).*

**2.1 Reviewer 2**

1. **Readability: A good illustration is the term SDRECS. While it is well defined in the paper, it is technical and not standard in similar literature. Why not simply write "change in outgoing radiation"? Most of the paper deals with shortwave under clear sky conditions, so this is implicit even from the title. The same goes for Rsutcsaf , Rsutcs and similar.**
The manuscript has been modified significantly to improve readability. Following both reviewers' recommendations most acronyms from the main text have been removed. There is no widely accepted acronym to designate the clear-sky shortwave direct aerosol effect. In figures, we have replaced SDRECS by $\mathrm{DRE_{clr}^{sw}}$, which we think is easier to understand. We have kept rsutcsaf and rsutcs notations as they are based on *CMIP6* naming convention (http://clipc-services.ceda.ac.uk/dreq/index/CMORvar.html). This is now clearly stated.

2. **One challenge, especially in the latter part of the paper (the regional trends and RF discussions), is to follow where the conclusions depend on the specific aerosol parametrizations of the GFDL model, and where they can be assumed to be more general. I would encourage the authors to add some further discussion of how model dependent the conclusions are. E.g. in the Conclusions, how general are the remarks about possible issues with the CEDS inventory? This is an important discussion for a dataset that will form the basis for much of CMIP6. A specific example: The authors conclude that "we find significant uncertainties in the CMIP6 emissions, including in the seasonality of NH3". In the paper, as far as I can understand, this is documented through the following: "We conducted a sensitivity simulation using the seasonality of NH3 column from AIRS (Warner et al., 2017) 265 to modulate NH3 emissions. We find that this revised seasonality significantly reduces the simulated winter trend in SDRECS (0.08 Wm − 2 dec − 1 ), improving the agreement with observations." I would expect some more discussion and documentation on this point, to make such a broad conclusion.**

   (a) We have added a figure in the supplementary materials comparing the seasonality of ammonia emissions modulated using CMIP6 seasonality and AIRS seasonality (see Fig. 1 below).

   (b) We have revised the conclusion to emphasize that some of the biases may be model specific while others are associated with CMIP6 emissions and will thus affect all models. The text was revised as follow:

   *Some of these biases may be model-specific, including the treatment of the mixing between sulfate and black carbon or the representation of the photochemistry of sulfate and nitrate. Others are attributed to the CMIP6 emissions and will likely affect other models. In particular, we find that the model bias in winter over India can be largely accounted for by uncertainties in the seasonality of ammonia and black carbon emissions. Similarly, comparisons between the CMIP6 and MEIC emission inventories over China suggest that the model bias in this region can be largely attributed to an underestimate of the reduction of $SO_2$ emissions after 2007.*

3. **For the DRF discussion, it would be good to put the results in a broader context. Aero-Com is mentioned; where is AM3 relative to the model mean in terms of forcing strengths? E.g. a comparison to the similar (but much less detailed) results in Myhre et al 2017, ACP ( https://www.atmos-chem-phys.net/17/2709/2017/) would be useful.**
We have added a comparison with the results from Myhre et al. [2017] in the RF section:

[Figure]

Figure 1: CMIP6 ammonia emissions for India with seasonality from CMIP6 (black) and AIRS (red)

*From 2001 to 2015, the direct aerosol forcing is simulated to be +0.03 $\mathrm{W\,m^{-2}}$, including +0.12, -0.03, and -0.03 $\mathrm{W\,m^{-2}}$ from black carbon, sulfate, and nitrate, respectively. Myhre et al. [2017] recently reported a similar change in the overall direct radiative forcing (+0.01 $\mathrm{W\,m^{-2}}$) but different contributions from sulfate (+0.03 $\mathrm{W\,m^{-2}}$) and black carbon (+0.03 $\mathrm{W\,m^{-2}}$). Many factors could contribute to these differences including the radiative properties of aerosols (e.g., the mixing of sulfate with black carbon [Bond et al., 2013]) and the emission inventories. Further studies are needed to examine whether changes in the sensitivity of radiative forcing to anthropogenic emissions are robust across models. Such assessment would be especially important in the northern midlatitudes, where the direct radiative forcing from aerosols and greenhouse gases from 2001 to 2015 are simulated to be of similar magnitude (+0.25 $\mathrm{W\,m^{-2}}$).*

4. **A more technical example: In Figure 1, introduced on line 177, the authors show both CEDS and MEIC emissions. However, "MEIC" isn't defined or discussed until line 300, making it difficult to understand even the first figure without already having read the entire manuscript. Please review for clarity, with a community reader in mind**
We have added the following text in the method section
*Anthropogenic emissions in India and China are expecte d to be more uncertain than in the US and Europe [Saikawa et al., 2017a,b]. Fig. 1 shows that the regional Modular Emission Inventory for China (MEIC) [Zhang et al., 2009], shows a decline of $SO_2$ emissions starting in 2006 and accelerating in 2012, a decrease of NO after 2012, and near-stable BC emissions after 2007. In 2014, MEIC NO, $SO_2$, and BC emissions are 24%, 48%, and 32% lower than CMIP6 emissions, respectively. $NH_3$ emissions are similar in magnitude but exhibit different seasonality: CMIP6 $NH_3$ emissions peak in spring, while MEIC exhibits a broad peak in summer, consistent with top-down constraints [Paulot et al., 2014, Zhang et al., 2017].*

[Figure]

**Figure 1.** Annual anthropogenic emissions of SO$_2$, BC, NH$_3$, and NO from  CMIP6 (solid lines) in selected regions. Emissions of SO$_2$, and NO with anthropogenic emissions from MEIC (for agriculture, energy, transportation, industry, and residential sectors) are also shown (dash lines).

[Figure]

**Figure 2.**  Decadal rate of change in the  clear-sky shortwave aerosol shortwave direct radiative effect ($DRE_{clr}^{sw}$). An increase in $DRE_{clr}^{sw}$ reflects a decrease in the amount of radiation scattered to space by aerosols. $EBAF_R$ is based on the outgoing clear-sky shortwave radiation  from CERES EBAF assuming its variability is solely associated with aerosols. $EBAF_C$ and  $EBAF_M$ are estimated using the observed clear-sky outgoing shortwave fluxes from CERES EBAF after accounting for the  variability of  water vapor, ozone,  and surface albedo from CERES-EBAF and MODIS, respectively.  Estimates from 23SYN (calculation constrained by observations) and from the GFDL AM3 global chemistry-climate model are also shown. Dotted areas are significant at the 95% confidence level.

[Figure]

**Figure 3.** Regional changes in the  clear-sky shortwave aerosol direct radiative effect derived from  CERES-EBAF outgoing radiation without correction ($\text{SDRECS}_{\text{CS}}$, $\text{EBAF}_{\text{R}}$ (black)) and after correcting for the variability of water, ozone, and surface albedo (from CERES-EBAF grey, andsurface albedoThe simulated annual anomaly in SDRECS$_{\text{AM3}}$ and in the outgoing shortwave radiation Rsutcplotted as -Rsutc for consistency with the definition of SDRECSredblue,magnitudethe linear decadal trend oftimeseries (in W m$^{-2}$ dec$^{-1}$)insetthe trend isat p=~~($p$<0.05).

[Figure]

**Figure 4.** Seasonal changes in the aerosol optical depth (AOD) and  clear-sky shortwave aerosol direct radiative effect (DRE$_{\mathrm{clr}}^{\mathrm{sw}}$) in Western Europe (Fig. 3). The top row shows the  AOD retrieved from different  spaceborne instruments (MODIS-Terra (lines), MODIS-Aqua (cross), MISR (diamond)) and the simulated  AOD decomposed into its components (bars). The second row shows the  simulated clear-sky shortwave aerosol  direct radiative effect of individual aerosols (bars) and the  overall aerosol direct radiative effect (white circle). The bottom  row shows  observation-based and simulated  estimates of changes in  the aerosol direct radiative effect.

[Figure]

**Figure 5.** Same as 4 for the Eastern US

[Figure]

**Figure 6.** Same as 4 for the India. MISR is excluded in the monsoon season, when its coverage is too sparse relative to MODIS (TERRA). The MODIS-derived dust optical depth is indicated by a black dash line.

[Figure]

**Figure 7.** Same as 4 for Eastern China. MISR is excluded in winter, spring, and monsoon seasons, when its coverage is too sparse.

same as Fig. 7 but with MEIC SO$_2$ and NO emissions, revised NH$_3$ seasonality, and heterogeneous oxidation of SO$_2$ (see text)

[Figure]

**Figure 8.** Meridional distribution of changes in anthropogenic emissions (BC, NO, NH$_3$, and SO$_2$) and in clear-sky (DRF$_{clr}$, middle row) and all-sky radiative aerosol direct radiative forcing (DRF, bottom row) from 1850 to 2001 (left) and from 2001 to 2015 (right). The thin black line indicates the instantaneous radiative forcing at TOA from well-mixed greenhouse gases. Global anthropogenic emissions and the total and speciated  DRF$_{clr}$ and DRF are indicated inline.

**Table 1.**  Trends in the aerosol optical depth (AOD,  $\mathrm{dec^{-1}}$), and direct clear-sky shortwave radiative effect ($\mathrm{DRE^{sw}_{clr}}$, $\mathrm{W\,m^{-2}\,dec^{-1}}$) for selected regions and seasons from 2002 to 2015[a]

[revised manuscript text omitted]

Vermote, E. F., El Saleous, N., Justice, C. O., Kaufman, Y. J., Privette, J. L., Remer, L., Roger, J. C., and Tanré,

985 D.: Atmospheric correction of visible to middle-infrared EOS-MODIS data over land surfaces: Background, operational algorithm and validation, Journal of Geophysical Research: Atmospheres, 102, 17 131–17 141, doi:10.1029/97JD00201, http://onlinelibrary.wiley.com/doi/10.1029/97JD00201/abstract, 1997.

Vermote, E. F., El Saleous, N. Z., and Justice, C. O.: Atmospheric correction of MODIS data in the visible to middle infrared: first results, Remote Sens. Environ., 83, 97–111, doi:10.1016/S0034-4257(02)00089-5,

990 http://www.sciencedirect.com/science/article/pii/S0034425702000895, 2002.

Wang, S., Xing, J., Jang, C., Zhu, Y., Fu, J. S., and Hao, J.: Impact Assessment of Ammonia Emissions on Inorganic Aerosols in East China Using Response Surface Modeling Technique, Environ. Sci. Technol., 45, 9293–9300, doi:10.1021/es2022347, http://dx.doi.org/10.1021/es2022347, 2011.

Wang, Y., Zhang, Q., Jiang, J., Zhou, W., Wang, B., He, K., Duan, F., Zhang, Q., Philip, S., and Xie, Y.:

995 Enhanced sulfate formation during China's severe winter haze episode in January 2013 missing from current models, J. Geophys. Res. Atmos., 119, 2013JD021 426, doi:10.1002/2013JD021426, http://onlinelibrary.wiley.com/doi/10.1002/2013JD021426/abstract, 2014a.

Wang, Z., Schaaf, C. B., Strahler, A. H., Chopping, M. J., Román, M. O., Shuai, Y., Woodcock, C. E., Hollinger, D. Y., and Fitzjarrald, D. R.: Evaluation of MODIS albedo product (MCD43A) over grassland, agricul-

1000 ture and forest surface types during dormant and snow-covered periods, Remote Sens. Environ., 140, 60–77, doi:10.1016/j.rse.2013.08.025, http://www.sciencedirect.com/science/article/pii/S0034425713002836, 2014b.

Warner, J. X., Dickerson, R. R., Wei, Z., Strow, L. L., Wang, Y., and Liang, Q.: Increased atmospheric ammonia over the world's major agricultural areas detected from space, Geophys. Res. Lett., p. 2016GL072305, doi:10.1002/2016GL072305, http://onlinelibrary.wiley.com/doi/10.1002/2016GL072305/abstract, 2017.

Wielicki, B. A., Barkstrom, B. R., Harrison, E. F., Lee, R. B., Louis Smith, G., and Cooper, J. E.: Clouds and the Earth's Radiant Energy System (CERES): An Earth Observing System Experiment, Bull. Amer. Meteor. Soc., 77, 853–868, doi:10.1175/1520-0477(1996)077<0853:CATERE>2.0.CO;2, http://journals. ametsoc.org/doi/abs/10.1175/1520-0477(1996)077%3C0853%3ACATERE%3E2.0.CO%3B2, 1996.

Wielicki, B. A., Barkstrom, B. R., Baum, B. A., Charlock, T. P., Green, R. N., Kratz, D. P., Lee, R. B., Minnis, P., Smith, G. L., Wong, T., Young, D. F., Cess, R. D., Coakley, J. A., Crommelynck, D. A. H., Donner, L., Kandel, R., King, M. D., Miller, A. J., Ramanathan, V., Randall, D. A., Stowe, L. L., and Welch, R. M.: Clouds and the Earth's Radiant Energy System (CERES): algorithm overview, IEEE Trans. Geosci. Remote Sens., 36, 1127–1141, doi:10.1109/36.701020, 1998.

Wild, M.: Global dimming and brightening: A review, Journal of Geophysical Research: Atmospheres, 114, D00D16, doi:10.1029/2008JD011470, http://onlinelibrary.wiley.com/doi/10.1029/2008JD011470/abstract, 2009.

Wu, Y., de Graaf, M., and Menenti, M.: Improved MODIS Dark Target aerosol optical depth algorithm over land: angular effect correction, Atmos. Meas. Tech., 9, 5575–5589, doi:10.5194/amt-9-5575-2016, https: //doi.org/10.5194/amt-9-5575-2016, 2016.

Xiao, Q., Zhang, H., Choi, M., Li, S., Kondragunta, S., Kim, J., Holben, B., Levy, R. C., and Liu, Y.: Evaluation of VIIRS,GOCI, and MODIS Collection 6 AOD retrievals against ground sunphotometer observations over East Asia, Atmos. Chem. Phys., 16, 1255–1269, doi:10.5194/acp-16-1255-2016, https://doi.org/10.5194/ acp-16-1255-2016, 2016.

Xing, J., Mathur, R., Pleim, J., Hogrefe, C., Gan, C.-M., Wong, D. C., and Wei, C.: Can a coupled meteorology–chemistry model reproduce the historical trend in aerosol direct radiative effects over the Northern Hemisphere?, Atmos. Chem. Phys., 15, 9997–10018, doi:10.5194/acp-15-9997-2015, https://www. atmos-chem-phys.net/15/9997/2015/, 2015.

Yevich, R. and Logan, J.: An assessment of biofuel use and burning of agricultural waste in the developing world., Global Biogeochem. Cycles, 17, 1095, 2003.

Zhang, L., Chen, Y., Zhao, Y., Henze, D. K., Zhu, L., Song, Y., Paulot, F., Liu, X., Pan, Y., and Huang, B.: Agricultural ammonia emissions in China: reconciling bottom-up and top-down estimates, Atmos. Chem. Phys. Discuss., 2017, 1–36, doi:10.5194/acp-2017-749, https://www.atmos-chem-phys-discuss.net/ acp-2017-749/, 2017.

Zhang, Q., Streets, D. G., Carmichael, G. R., He, K. B., Huo, H., Kannari, A., Klimont, Z., Park, I. S., Reddy, S., Fu, J. S., Chen, D., Duan, L., Lei, Y., Wang, L. T., and Yao, Z. L.: Asian emissions in 2006 for the NASA INTEX-B mission, Atmos. Chem. Phys., 9, 5131–5153, doi:10.5194/acp-9-5131-2009, https://www. atmos-chem-phys.net/9/5131/2009/, 2009.

Zhao, B., Jiang, J. H., Gu, Y., Diner, D., Worden, J., Liou, K.-N., Su, H., Xing, J., Michael Garay, and Huang, L.: Decadal-scale trends in regional aerosol particle properties and their linkage to emission changes, Environ. Res. Lett., 12, 054 021, doi:10.1088/1748-9326/aa6cb2, http://stacks.iop.org/1748-9326/12/i=5/a=054021, 2017.

Zheng, B., Zhang, Q., Zhang, Y., He, K. B., Wang, K., Zheng, G. J., Duan, F. K., Ma, Y. L., and Kimoto, T.: Heterogeneous chemistry: a mechanism missing in current models to explain secondary inorganic aerosol formation during the January 2013 haze episode in North China, Atmos. Chem. Phys., 15, 2031–2049, doi:10.5194/acp-15-2031-2015, http://www.atmos-chem-phys.net/15/2031/2015/, 2015.

Zhu, Z., Piao, S., Myneni, R. B., Huang, M., Zeng, Z., Canadell, J. G., Ciais, P., Sitch, S., Friedlingstein, P., Arneth, A., Cao, C., Cheng, L., Kato, E., Koven, C., Li, Y., Lian, X., Liu, Y., Liu, R., Mao, J., Pan, Y., Peng, S., Peñuelas, J., Poulter, B., Pugh, T. A. M., Stocker, B. D., Viovy, N., Wang, X., Wang, Y., Xiao, Z., Yang, H., Zaehle, S., and Zeng, N.: Greening of the Earth and its drivers, Nature Climate Change, 6, 791–795, doi:10.1038/nclimate3004, https://doi.org/10.1038/nclimate3004, 2016.

---

## Author Response (AR2)

We thank both reviewers for their comments. Replies and changes are listed below.

**1 Reviewer 1**

**1. There needs to be a description of the errors in the AOD comparisons and how these project onto the attribution of the AOD changes. Currently, the approach appears to be the AODs and its trends look about the same, therefore they agree, however without any description of the errors it is impossible to tell if this conclusion is robust.**

We agree with the reviewer. We have modified the manuscript to make it clear whether differences between model and observations are robust. Specifically, we have also added a Table in the supplementary materials (Table S2), which provides the confidence intervals for AOD (and DREclrsw) trends listed in Table 1. Table S2 supports our assertion that AM3 can capture changes in AOD well (i.e., falls within the uncertainty of the observed trends) in Europe and in the US (spring) but exhibits significant biases in India and China. For completeness, we have also added the uncertainty in MODIS and MISR individual retrievals. (see reply to next comment). We also wish to clarify that the model aerosol simulation (including AOD) has been extensively validated against observations in a previous study [Paulot et al., 2016]. This has been clarified in the text:

The configuration of AM3 used in this study includes revisions to the representation of the wet scavenging of chemical tracers by snow and convective precipitation and to the treatment of sulfate and nitrate chemistry, which significantly improve the representation of aerosols. We refer the reader to our recent work for a detailed evaluation of the aerosol simulation in AM3 [Paulot et al., 2016].

**2. There is also no discussion on how these errors as well as errors in the interferences (e.g. albedo and its changes) may affect the projection of AOD variability to forcing variability.**

The new Table S2 shows how uncertainties in AOD trends propagate to uncertainties in DRE trends. It is clear from our study and others, that there is no universal relationship between AOD and DRE. In particular, this relationship depends on aerosol and surface properties, as highlighted by the reviewer. We have now clarified how these processes affect the interpretation of differences between observed and simulated trends in  $DRE_{sw}^{clr}$  in the method section. The following text was added to the section devoted to the trend calculation (renamed: *Trend: estimation and interpretation*).

Differences between observed and simulated trends in  $DRE_{clr}^{sw}$  may reflect biases in the simulated change of the aerosol burden. Here this is diagnosed by comparing the simulated trend in aerosol optical depth (AOD) with those retrieved by the Multi-angle Imaging SpectroRadiometer (MISR) at 555nm [Kahn et al., 2005, 2010] and the MODIS instruments on board the AQUA and TERRA satellites at 550 nm (collection 6, level3, merged deep blue/dark target) [Levy et al., 2013, Sayer et al., 2014]. Note that the accuracy of individual retrievals has been estimated to be  $\pm 0.05 \pm 0.15 \times AOD$  [Levy et al., 2010] for MODIS and the maximum of  $\pm 0.05$  or  $0.2 \times AOD$  for MISR [Kahn et al., 2010]. The change in AOD is not a perfect predictor of changes in DRE\_sw and we will show that it is possible to find regions where observed changes in AOD are well captured by AM3 but not changes in DRE\_sw (see 3.2.2). Such discrepancies may reflect differences in aerosol radiative properties. Specifically, changes in absorbing aerosols, such as black carbon, have a small imprint on AOD but a large one on DRE\_sw trends. For instance, a lower surface albedo reduces the impact of changes in scattering aerosols on DRE\_sw and conversely increases that of absorbing aerosols. We will show that such differences in surface albedo are important in India (section 3.2.2). However, in other regions, we find such differences have a small impact on the simulated trend in DRE\_sw.

The importance of aerosol properties has been further highlighted (in addition to the existing discussion regarding India) in the section devoted to China as follows: Similar to India (in winter), the discrepancy between the model performances for AOD and  $DRE_{sw}^{clr}$  points to a bias in aerosol properties. In particular, MEIC suggests that BC emissions have remained stable from 2005 up to 2013. If instead BC emissions

increased over this time period as suggested by the historical CMIP6 emissions, the change in  $DRE_{clr}^{sw}$  would be reduced without significant impact on the simulated AOD.

As far as observations are concerned, our derivation of  $DRE_{clr}^{sw}$  (  $EBAF_C$  and  $EBAF_M$  ) does not use AOD, unlike the SYN product and we consider two different albedo retrievals (MODIS for  $EBAF_M$  and CERES-EBAF for  $EBAF_C$ ) to characterize the impact of errors in the albedo retrievals on our conclusions.

3. There is also a substantial issue with the model in the tropics and its unclear how this structural error projects into decadal and centennial variations in the forcing.

We have addressed the reviewer's comment regarding the treatment of biomass burning in AM3 in the previous round of reviews. Without more details it is unclear which *structural uncertainty* in the model the reviewer is referring to. We have modified the text to emphasize that the version of AM3 used here includes modification to the wet deposition treatment that improve the representation of aerosols, especially in the tropics [Paulot et al., 2016].

The configuration of AM3 used in this study includes revisions to the representation of the wet scavenging of chemical tracers by snow and convective precipitation and to the treatment of sulfate and nitrate chemistry, which significantly improve the representation of aerosols [Paulot et al., 2016].

- 4. Section 2.2.How do you account for differences in AOD observed by MISR and MODIS? We have added the uncertainty in both MODIS and MISR retrievals. While there are differences between these retrievals, there is excellent agreement regarding regarding the magnitude of the trends (similar to Zhao et al. [2017]) as shown in Table 1 and S2. Further discussion of differences in the MODIS and MISR retrieval algorithms are beyond the scope of this study.
- 5. Equation 1: Please rename these acronyms (rsutcsaf, rustics). Its not obvious to a reader that these should be related to outgoing clear-sky shortwave radiation, especially as a reader attempts to keep track of these (and other acronyms) throughout the paper.

To our knowledge, there is no standard acronym to designate outgoing clear-sky shortwave radiation and outgoing clear-sky shortwave radiation in the absence of aerosols. Therefore, we have decided to adopt the naming convention used by CMIP6. The text has been revised as follows:

where we use the CMIP6 convention [CMIP6 Data Request, 2018] to designate the outgoing clear-sky shortwave radiation with (rsutcs) and without aerosols (rsutcsaf), respectively. For simplicity, we will refer to the aerosol shortwave direct radiative effect under clear-sky conditions (DREswclr) as the aerosol effect, hereafter.

6. Line 145: What is this?: EBAFR hereafter (R:raw)? Which estimate are you referring too? EBAFR is referring to the change in the outgoing clear-sky shortwave radiation as derived from the CERES-EBAF product. We have revised the text as follows:

The simplest way in which CERES EBAF data can be used to estimate changes in the aerosol effect is to assume that all variability in the shortwave clear-sky outgoing radiation is the result of changes in aerosols [Stevens and Schwartz, 2012, Xing et al., 2015, Alfaro-Contreras et al., 2017]. We will refer to this estimate as  $EBAF_R$  hereafter, where R stands for raw.

- 7. Line 195: What is a dec-1, decade-1? Correct. We have replaced  $dec^{-1}$  by  $decade^{-1}$ .
- 8. Why is there a section on aerosol optical depth (from MISR and MODIS) and then another on observations from CERES?

The discussion of the AOD has been moved into section 2.3, where we now discuss how we use AOD to interpret changes in  $DRE_{sw}^{clr}$ . See replies to comments 1 and 2.

9. Line 145 and supplemental, there is insufficient information on how you estimate the change in outgoing radiation after accepting for changes in albedo.

The approach used here to estimate the impact of changes in surface albedo, water vapor, and ozone on the outgoing radiation is widely used [Soden et al., 2008, Shell et al., 2008] and we show that it works well in Fig. S1. We have revised the method section as follows:

Therefore, a more accurate estimate of the aerosol effect requires removal of the impact of these components from the measured changes in the outgoing radiation. To achieve this, we calculate radiative kernels (e.g., Soden et al. [2008], Shell et al. [2008]) to estimate the variability of the outgoing clear-sky shortwave radiation associated with changes in surface albedo, ozone, and water vapor (see supporting materials and Fig. S1).

and we have added the following text to the supplementary materials:

We calculate radiative kernels [Soden et al., 2008, Shell et al., 2008] to estimate the change in aerosol-free clear-sky outgoing shortwave radiation (rsutcsaf) due to perturbations in surface albedo (salb), water vapor (WVP), and ozone (qo3).

We evaluate our methodology by comparing the annual variability of rsutcsaf calculated in CERES-SYN over the 2001-2015 period with that estimated using the radiative kernels introduced above. CERES-SYN rsutcsaf is calculated using surface albedo, water vapor, and ozone constrained by observations inputted into a radiative transfer code. We use the same surface albedo, water vapor, and ozone in conjunction with our Jacobian to estimate the variability in rsutcsaf.

10. Also, the variations over the oceans do not appear to follow any type of outflow pattern; are these related to changes in clouds?

We agree with the reviewers that some significant changes are diagnosed over remote regions. This may reflect cloud contaminations in the CERES cloud filtering algorithm (for  $EBAF_C$ ,  $EBAF_M$ ) and in the aerosol retrieval (SYN). In addition, the derivation of the DRE under very low aerosol loadings will be more susceptible to errors in the radiative kernels.

We have added the following text to the Global distribution of changes in aerosol effect subsection:

We note that all observation-based estimates of DRE show some significant changes over remote oceanic regions. These changes may reflect cloud contaminations in the CERES cloud filtering algorithm (for  $EBAF_{C}$ ,  $EBAF_{M}$ ) and in the aerosol retrieval (SYN). In addition, low aerosol loadings make  $EBAF_{C}$ ,  $EBAF_{M}$  more susceptible to errors in the radiative kernels.

**11. The SYN product appears to be much cleaner relative to the EBAF products (variations over the ocean could be due to outflow of aerosols as opposed to cloud variability). Why not use this product for your evaluation?**

The SYN product is calculated using observational constraints on the aerosols inferred from the MATCH model. Hence, the SYN product is expected to be *smoother* than estimates derived from EBAF observations. However, it is not directly related to the observed change in the outgoing radiation. In addition SYN is based on MODIS collection 5, which exhibits significant differences from the newer (and recommended) collection 6. We think it is also important to provide different estimates of the change in DRE to help quantify the observational uncertainty (similar to AOD).

**12. Paper structure**

We appreciate the reviewer' suggestions to split the paper into two separate studies. However, we think that it is essential to provide the reader with both the model evaluation and the simulated aerosol forcing in a single manuscript. In particular, our estimate of the change in the aerosol direct forcing makes use of revisions to the anthropogenic emissions over India and China that are based on the evaluation of the model against observations. In addition, we note that we have already evaluated the model aerosol simulation in a previous study [Paulot et al., 2016] and we have reorganized the method section to clarify that AOD is used in our study to interpret changes in  $DRE_{sw}^{clr}$ .

**2 Reviewer 2**

1. The paper consistently refers to the Hoesly et al 2018 emission dataset as the CMIP6 emissions, and correctly note that they don't reflect the recent reduction in SO2 emissions from China. It is my understanding however that CEDS will make a last minute change to their emission to take this into account. Hence I fear that there will be some confusion in the litterature as to what "CMIP6 emissions" are. I'm not sure what the best solution would be, but think it might be prudent to refer to the anthropogenic emissions as CEDS [Hoesly 2018]) or similar, rather than CMIP6.

Historical CEDS emissions, which are used in this manuscript, are frozen. The reviewer is correct that changes in CEDS emissions may still occur but such changes will only affect emissions from 2015 onward. We have revised the text as follows:

We use the historical anthropogenic emissions developed by the Community Emission Data System (CEDS v2017-05-18) in support CMIP6 [Hoesly et al., 2018].

- 2. **148:** Missing colon behind "as follows" Thank you, this has been corrected
- 3. Figure 8, panel e: The global mean hides the peak in the instantaneous GHG forcing line. Thank you, this has been corrected

Table S2: Observation-based and simulated decadal trends for the AOD and direct clear-sky shortwave radiative effect ( $DRE_{clr}^{sw}$ ,  $Wm^{-2} decade^{-1}$ ) for selected regions and seasons from 2002 to  $2015^a$

|                              | Western Europe                    |                     | Eastern US          |                     | India             |                   | Eastern China     |
|------------------------------|-----------------------------------|---------------------|---------------------|---------------------|-------------------|-------------------|-------------------|
|                              | MAM                               | JJA                 | MAM                 | JJA                 | DJF               | MAM               | MAM               |
| AOD                          |                                   |                     |                     |                     |                   |                   |                   |
| MODIS (TERRA)                | $-0.04 \left[-0.05, -0.02\right]$ | -0.04 [-0.07,-0.02] | -0.04 [-0.06,-0.01] | -0.11 [-0.16,-0.09] | 0.13 [0.10, 0.16] | 0.04 [0.00, 0.07] | * [-0.04,0.07]    |
| MODIS (AQUA)                 | -0.05 [-0.06,-0.02]               | -0.03 [-0.05,-0.02] | -0.04 [-0.06,-0.01] | -0.10 [-0.17,-0.07] | 0.11 [0.08,0.15]  | 0.07 [0.03, 0.09] | * [-0.08,0.08]    |
| MISR                         | -0.03 [-0.04,-0.02]               | -0.03 [-0.06,-0.02] | -0.02 [-0.04,-0.01] | -0.08 [-0.12,-0.07] | 0.05 [0.03,0.07]  | * [0.01,0.05]     |                   |
| MATCH                        | -0.06 [-0.07,-0.04]               | -0.06 [-0.10,-0.05] | -0.07 [-0.10,-0.05] | -0.11 [-0.15,-0.10] | 0.10 [0.07,0.13]  | 0.03 [0.01, 0.06] | * [-0.07,0.12]    |
| AM3                          | -0.04 [-0.05,-0.01]               | -0.05 [-0.08,-0.03] | -0.03 [-0.05,-0.01] | -0.05 [-0.07,-0.04] | 0.13 [0.10, 0.17] | 0.15[0.11, 0.18]  | 0.15 [0.12, 0.22] |
| $DRE_{clr}^{sw}$             |                                   |                     |                     |                     |                   |                   |                   |
| SYN                          | 1.8 [1.3, 2.2]                    | 2.5 [2.0, 3.2]      | 2.1 [1.3, 2.7]      | 3.6[3.0, 4.4]       | -2.6 [-3.3,-2.1]  | -1.4 [-2.2, -0.6] | * [-1.7,1.5]      |
| $\mathrm{EBAF}_{\mathrm{C}}$ | $1.4 \ [0.6, 1.6]$                | 1.8 [1.5, 2.4]      | 1.3 [0.5, 2.2]      | 3.3 [2.7, 4.2]      | -2.3 [-3.2,-2.1]  | -1.2 [-2.2, -0.6] | * [-1.7,0.8]      |
| $\mathrm{EBAF}_{\mathrm{M}}$ | $1.0 \ [0.3, 1.3]$                | 1.2 [1.0, 1.7]      | * [-0.0,1.5]        | 2.0 [1.1, 3.1]      | -0.8 [-1.6,-0.6]  | -0.9 [-1.2, -0.4] | * [-1.4,0.7]      |
| AM3                          | $1.1 \ [0.3, 1.2]$                | 1.5 [0.9, 2.2]      | $0.9 \ [0.5, 1.2]$  | 1.4 [1.2, 2.0]      | -2.7 [-3.2,-2.2]  | -3.1 [-3.9, -2.6] | -2.1 [-3.5,-1.6]  |

a The trend is estimated using the Theil-Sen method. Bootstrap estimates of the 95% confidence interval are indicated in bracket. \* denote non significant monotonous change at p=0.05

Bin Zhao, Jonathan H. Jiang, Yu Gu, David Diner, John Worden, Kuo-Nan Liou, Hui Su, Jia Xing, Michael Garay, and Lei Huang. Decadal-scale trends in regional aerosol particle properties and their linkage to emission changes. *Environ. Res. Lett.*, 12(5):054021, 2017. ISSN 1748-9326. doi: 10.1088/1748-9326/aa6cb2. URL http://stacks.iop.org/1748-9326/12/i=5/a=054021.

Manuscript prepared for Atmos. Chem. Phys. with version 2014/09/16 7.15 Copernicus papers of the LATEX class copernicus.cls. Date: 20 July 2018

**Changes in the aerosol direct radiative forcing from 2001 to 2015: observational constraints and regional mechanisms**

Fabien Paulot1,2, David Paynter1, Paul Ginoux1, Vaishali Naik1, and Larry W. Horowitz1

1Geophysical Fluid Dynamics Laboratory, National Oceanic and Atmospheric Administration, Princeton, New Jersey, USA

2Program in Atmospheric and Oceanic Sciences, Princeton University, New Jersey, USA

Correspondence to: Fabien.Paulot@noaa.gov

Abstract. We present observation- and model-based estimates of changes in the aerosol direct clearsky shortwave radiative effect ( $DRE_{clr}^{sw}$ ), the perturbation by aerosols of the net downward shortwave clear-sky radiation at the top of the atmosphere. Observation-based estimates of  $DRE_{clr}^{sw}$  are derived from the outgoing shortwave clear-sky radiation measured by the Clouds and the Earth's Radiant

- 5 Energy System (CERES) accounting for the effect of variability in surface albedo, water vapor, and ozone. From 2001 to 2015, we find that  $DRE_{clr}^{sw}$  increases (i.e., less radiation is scattered to space by aerosols) over Western Europe (0.7 1  $Wm^{-2} dec^{-1}Wm^{-2} dec^{-1}$ ) and the Eastern US (0.9 1.8  $Wm^{-2} dec^{-1}Wm^{-2} decade^{-1}$ ), decreases over India (-0.5 -1.9  $Wm^{-2} dec^{-1}Wm^{-2} 
[revised manuscript text omitted]

---

## Author Response (AR3)

We thank the reviewer and the editor for their comments.

1. **please consider the referee's comment regarding your abstract. I generally agree that the abstract could represent the findings of the paper more simply and clearly. For example, I actually found your existing conclusions to be a clearer statement of the main points of the manuscript.**

   We have revised the abstract as follow:

   *We present estimates of changes in the direct aerosol effects (DRE) and its anthropogenic component (DRF) from 2001 to 2015 using the GFDL chemistry-climate model AM3 driven by CMIP6 historical emissions. AM3 is evaluated against observed changes in the clear-sky shortwave direct aerosol effect ($DRE_{sw}^{clr}$) derived from the Clouds and the Earth's Radiant Energy System (CERES) over polluted regions. From 2001 to 2015, observations suggest that $DRE_{clr}^{sw}$ increases (i.e., less radiation is scattered to space by aerosols) over Western Europe (0.7 – 1 $W\,m^{-2}\,decade^{-1}$) and the Eastern US (0.9 – 1.8 $W\,m^{-2}\,decade^{-1}$), decreases over India (-0.5 – -1.9 $W\,m^{-2}\,decade^{-1}$), and does not change significantly over Eastern China. AM3 captures these observed regional changes in $DRE_{clr}^{sw}$ well in the US and Western Europe, where they are dominated by the decline of sulfate aerosols, but not in Asia, where the model overestimates the decrease of $DRE_{clr}^{sw}$. Over India, the model bias can be partly attributed to a decrease of dust optical depth, which is not captured by our model and offsets some of the increase of anthropogenic aerosols. Over China, we find that the decline of $SO_2$ emissions after 2007 is not represented in the CMIP6 emission inventory. Accounting for this decline using the Modular Emission Inventory for China and for the heterogeneous oxidation of $SO_2$ significantly reduce the model bias. For both India and China, our simulations indicate that nitrate and black carbon contribute more to changes in $DRE_{clr}^{sw}$ than in the US and Europe. Indeed, our model suggests that black carbon (+0.12 $W\,m^{-2}$) dominates the relatively weak change in DRF from 2001 to 2015 (+0.03 $W\,m^{-2}$). Over this period, the change in the forcing from nitrate and sulfate are both small and of the same magnitude (-0.03 $W\,m^{-2}$ each). This is in sharp contrast to the forcing from 1850 to 2001 in which forcings by sulfate and black carbon largely cancel each others, with minor contributions from nitrate. The differences between these time periods can be well understood from changes in emissions alone for black carbon but not for nitrate and sulfate, which reflects non-linear changes in their photochemical production associated with changes in both the magnitude and spatial distribution of anthropogenic emissions.*

2. **On line 116, I don't understand the origin of the rsutcs and rsutcaf acronyms. Please either use acronyms reflecting the descriptive terms in your text or, if those are common acronyms in the subfield, please state what they stand for. Not knowing what they stand for makes it difficult to follow discussion later when the come back up.**

   rsutcsaf and rsutcs are the variable names adopted by the Coupled Model Intercomparison Project (6). We have spelled out these acronyms in the revised version of the manuscript.

   *the outgoing clear-sky shortwave radiation with (rsutcs: radiation shortwave up toa clear sky) and without aerosols (rsutcsaf: radiation shortwave up toa clear sky aerosol free), respectively*

3. **In section 2.2.2, I think it might help non-modelers and people not well versed in radiative transfer calculations to explicitly discuss the differences between the EBAF and SYN products. You do describe the SYN product but I have a hard time evaluating what agreement or disagreement with one or the other might mean since I don't have a great sense for how they relate to each other.**

   CERES SYN provides estimates of the aerosol effect using a radiative calculation performed with and without aerosols similar to AM3. In CERES-SYN, aerosol properties are obtained from the MATCH model, which is constrained by observations from MODIS. The text was revised as follow:

   *We also derive the change in the aerosol effect from the CERES Synoptic Radiative Fluxes product (SYN, edition 4a). Similar to AM3, the CERES SYN product provides estimates of the radiative fluxes at the top of the atmosphere with and without aerosols present. In SYN, the radiative transfer calculations use aerosol properties from the Model for Atmospheric Transport and Chemistry (MATCH), which is constrained by observations from MODIS collection 5 [Collins et al., 2001].*

4. **Line 156, recommend rewording "(MATCH) that" to "(MATCH), which"**

   corrected

5. **Line 165, recommend rewording "environmental dataset" to "an environmental dataset"**

   corrected

6. **I found lines 404-405 confusing. DRF for both 2001 and 2015 is already relative to 1850 correct? So is this sentence simply saying that DRF due to BC decreased by 75% from 2001 to 2015? The following sentence regarding sulfate is clearer. Please reword.**

   No it means that the forcing has increased by 25%. We have revised the text as follow.

   *The clear-sky direct radiative forcing of black carbon increases by 25% from 2001 to 2015, in good agreement with the change in BC emissions.*

**References**

William D. Collins, Phillip J. Rasch, Brian E. Eaton, Boris V. Khattatov, Jean-Francois Lamarque, and Charles S. Zender. 
[revised manuscript text omitted]